Pathogens

# Neutralization of SARS-CoV-2 by IgM-14 via engagement of two distinct spike epitopes

Yan Wang[1☉], Yanping Hu[1☉], Zhiqiang Ku[2], Jason Yeung[3], Jing Zou[1], Michael Woodson[3], Nikolai S. Prokhorov[3], Ekaterina S. Knyazhanskaya[3], Haiqing Zhao[3], Michael B. Sherman[3], Zhiqiang An[2], Stephen F. Carroll[4], Pei-Yong Shi[3], Petr G. Leiman[3*], Xuping Xie[1,5*]

1 Department of Microbiology and Immunology, University of Texas Medical Branch, Galveston, Texas, United States of America, 2 Texas Therapeutics Institute, Brown Foundation Institute of Molecular Medicine, The University of Texas Health Science Center at Houston, Houston, Texas, United States of America, 3 Department of Biochemistry and Molecular Biology, University of Texas Medical Branch, Galveston, Texas, United States of America, 4 IGM Biosciences Inc., 325 East Middlefield Road, Mountain View, California, United States of America, 5 Sealy Institute for Drug Discovery, University of Texas Medical Branch, Galveston, Texas, United States of America

☉ These authors contributed equally to this study
* pgleiman@utmb.edu (PGM); xuxie@utmb.edu (XX)

## Abstract

Engineered immunoglobulin M (IgM) antibodies typically exhibit superior neutralization potency and avidity compared to their parental IgG counterparts, primarily due to multivalent binding to repeated epitopes on a targeting antigen. In this study, we characterize the neutralization breadth and mechanism of action of IgM-14, a previously reported intranasally deliverable antibody targeting SARS-CoV-2. IgM-14 demonstrates remarkably potent antiviral activity against all pre-Omicron variants but significantly reduced efficacy against Omicron BA.1, and complete loss of activity against the later subvariant JN.1. Resistance selection identified two key mutations in the receptor-binding domain (RBD), G476D and F486P, which disrupt IgM-14 binding and confer strong resistance. Cryo-electron microscopy analysis uncovered two distinct Fab-RBD interfaces: a primary interface overlapping the angiotensin-converting enzyme 2 (ACE2)-binding region, and a unique secondary interface formed only when the RBD adopts the ACE2-inaccessible "down" conformation, involving a neighboring spike protomer. Site-directed mutagenesis and structural modeling revealed a critical role of this secondary site in IgM-14-mediated neutralization. Unlike IgG-14, structural modeling suggested that IgM-14 can simultaneously engage both interfaces in diverse modes, indicating a noncanonical avidity mechanism. Collectively, these findings highlight the structural and functional uniqueness of IgM-14 and offer valuable insights into the rational design of next-generation spike-targeted antibody therapeutics with enhanced breadth and potency.

**Data availability statement:** The cryo-EM maps and atomic models have been deposited in the Electron Microscopy Data Bank (EMDB) and the Protein Data Bank (PDB) under the following accession codes: apo D614G spike has accession codes EMD-73231 and EMD-73244; Fab-14/D614G spike complex has accession codes EMD-73260, EMD-73228, EMD-73263, EMD-73265, EMD-73267, EMD-73290, EMD-73306, EMD-73245, EMD-73292, EMD-73247, EMD-73291, PDB:9YOK, PDB:9YNR, PDB:9YPR, PDB:9YNX, PDB:9YPB. Fab-14/Omicron BA.1 spike complex has accession codes EMD-73270, EMD-73273, and EMD-73271. All relevant data are within the manuscript and its Supporting Information files.

**Funding:** X.X. were supported by awards from the Sealy Smith Foundation, Robert J. Kleberg, Jr. and Helen C. Kleberg Foundation, John S. Dunn Foundation, Amon G. Carter Foundation, Gillson Longenbaugh Foundation, and Summerfield Robert Foundation. P.G.L. is supported by the NIGMS grants R01GM139034 and R35GM158090. P.-Y.S., X.X., and P.G.L.'s team received research funding from IGM Biosciences Inc. P.G.L., N.S.P., and P.-Y.S. were awarded a COVID-19 Pilot grant by the UTMB Institute for Human Infections and Immunity (IHII). The funders had no role in study design, data collection and analysis, decision to publish, or preparation of the manuscript.

**Competing interests:** I have read the journal's policy and the authors of this manuscript have the following competing interests: the University of Texas System has filed a patent on the SARS-CoV-2 antibodies and the reverse genetic system, and reporter SARS-CoV-2. The University of Texas System and IGM Biosciences Inc. have filed a joint patent on the SARS-CoV-2 IgM antibodies. S.F.C. was an employee of IGM Biosciences, Inc. during this study. Other authors declare no competing interests.

## Author summary

Antibody therapies are essential for COVID-19 prevention and treatment. However, many traditional IgG-based antibodies have become less effective as SARS-CoV-2 continues to evolve. Engineered IgM antibodies offer advantages because they can bind to viruses at multiple sites simultaneously, increasing both strength and breadth of protection. In this study, we investigated IgM-14, an engineered IgM antibody designed for intranasal delivery. We found that IgM-14 strongly neutralizes early SARS-CoV-2 variants but loses effectiveness against Omicron lineages, including complete inactivation against the recent JN.1 variant. We also identified two key spike mutations that enable the virus to escape IgM-14. To understand the basis of IgM-14's activity, we used cryo-electron microscopy to examine how it binds to the viral spike protein. Surprisingly, the antigen-binding fragment of IgM-14 engages two distinct sites on the receptor-binding domain: a primary site that overlaps that of the ACE2 receptor, and a second site that appears only when the spike is in the "down" position. Structural modeling and mutational analysis reveal that this secondary site contributes to IgM-14's unusually strong neutralizing ability. Our work highlights a previously underappreciated mechanism by which engineered IgM antibodies achieve high potency and offer insights for designing next-generation antibodies that better resist viral evolution.

## Introduction

The Coronavirus Disease 2019 (COVID-19) pandemic, caused by severe acute respiratory syndrome coronavirus 2 (SARS-CoV-2), has precipitated a global health and economic crisis, leading to millions of deaths and widespread morbidity [1]. Since the early pandemic, vaccines and monoclonal antibodies have been developed at unprecedented speed, with many deployed to treat COVID-19 patients. However, the emergence of SARS-CoV-2 variants with increased immune evasion and transmissibility has posed significant challenges to antibody-based therapies. Variants such as Omicron and its sublineages, such as BA.1, BA.2, BA.4/5, and BA.2.86/JN.1, harbor many mutations that weaken the effectiveness of existing vaccines and monoclonal antibodies [2–4]. Consequently, several clinically approved monoclonal antibodies have been withdrawn due to their loss of potency against these evolving variants [5]. Continuous updates on vaccines and monoclonal antibodies are essential to protect vulnerable populations from newly emerged SARS-CoV-2 variants.

SARS-CoV-2 belongs to the genus Betacoronavirus, subgenus *sarbecovirus* lineage B [6]. Similar to other coronaviruses, SARS-CoV-2 possesses a homotrimeric spike (S) glycoprotein that protrudes from the lipid envelope of the virus and forms a crown-like structure (so-called "corona"). The spike mediates viral attachment and entry into host cells and determines the virus's host range and tissue tropism [7,8]. Following biosynthesis, most SARS-CoV-2 spikes are cleaved into S1 and

S2 subunits by host proteases, including furin. The S1 subunit contains two large domains: an N-terminal domain (NTD; residues 14–303) and a receptor-binding domain (RBD; residues 319–541) [7], with the latter being responsible for viral attachment to its primary receptor, angiotensin-converting enzyme 2 (ACE2). In the spike structure, there are two most populous conformations of the RBD: "up", ACE2-accessible; and "down", ACE2-inaccessible [7]. The S2 subunit mediates fusion of viral and host cell membranes and contains a fusion peptide (FP), heptapeptide repeat sequences (HR1 and HR2), a transmembrane domain, and a C-terminal cytoplasmic tail [9].

The spike protein is the primary target of antibody-based therapeutics. Neutralizing antibody (NAb) inhibits viral attachment and/or fusion by binding to specific spike epitopes. Most potent SARS-CoV-2 NAbs identified to date target the RBD and can be categorized into four classes based on their epitopes [10]. Class 1 antibodies bind the epitopes within the receptor-binding motif (RBM) when the RBD is in the up conformation. Class 2 antibodies recognize RBM epitopes that are accessible in both conformations. Class 3 antibodies target regions outside the RBM, regardless of the RBD's up or down conformations. Class 4 antibodies bind conserved cryptic regions within the RBD. Class 1 and 2 antibodies potently inhibit SARS-CoV-2 entry by directly competing with ACE2. In contrast, most class 3 and 4 antibodies typically neutralize the virus indirectly by stabilizing the spike conformation, introducing steric hindrance between the RBD and ACE2, preventing spike conformational transitions, or blocking membrane fusion [10]. However, SARS-CoV-2 variants have evolved multiple mechanisms to evade NAb-mediated neutralization, including mutations at epitope positions, altered glycan shielding, and changes in spike conformation and flexibility [4,11].

Our previous research identified an immunoglobulin G (IgG-14) from an antibody phage display library that potently neutralized SARS-CoV-2 parental strains [12]. IgG-14 was subsequently engineered into a nasally deliverable immunoglobulin M (IgM-14), which demonstrated favorable pharmacokinetics, extremely high potency, and a much broader antiviral spectrum against pre-Omicron SARS-CoV-2 variants relative to the parental IgG [13]. These attributes facilitated the progression of IgM-14 to clinical trials [14]. However, its clinical development was discontinued due to a marked reduction in efficacy against Omicron variants, a challenge encountered by many antibodies targeting pre-Omicron variants [2–4]. Despite these setbacks, understanding the molecular mechanisms underlying IgM-14's superior properties over IgG-14 and the evasion mechanism by Omicron variants should greatly facilitate the rational design of future IgM-based therapies to combat SARS-CoV-2 variants and other coronaviruses.

In this study, we systematically evaluated the neutralizing activities of IgM-14 and IgG-14 against a broad range of SARS-CoV-2 variants, including a later Omicron subvariant JN.1. We also examined IgM-14's potency against Omicron BA.1 in physiologically relevant human primary airway epithelial cell cultures. Using an *in vitro* resistance selection approach, we identified spike protein point mutations that disrupt antibody binding and confer resistance to IgM-14. Furthermore, we solved high-resolution structures of the SARS-CoV-2 spike protein in complex with the fragment antigen-binding region of IgM-14 (Fab-14). Structural and molecular analysis revealed distinct binding modes and unique mechanisms that underlie IgM-14's enhanced neutralization potency and breadth.

## Results

### Neutralization of IgM-14 against naturally emerged SARS-CoV-2 variants

Human IgM-14 was previously engineered from its parental IgG-14 by replacing the Fv region of IgG1 with an IgM scaffold in the presence of J-chain co-expression [13]. Unlike IgG-14, which contains two identical Fabs, IgM-14 features ten Fabs (Fig 1A), enhancing its valency and avidity. Accordingly, IgM-14 outperformed IgG-14 in the neutralization of multiple SARS-CoV-2 variants, including B.1.1.7 (Alpha), B.1.351 (Beta), P.1 (Gamma), and even the spike E484A mutant, which is resistant to IgG-14 [12,13].

In this study, we further evaluated IgM-14's neutralization activity against a range of additional SARS-CoV-2 variants, including pre-Omicron variants-D614G, B.1525 (Eta), B.1.526 (Iota), B.1.617.1 (Kappa), B.1.617.2 (Delta), C.37 (Lambda), B.1.618, B.1.621 (Mu), as well as highly mutated Omicron sublineages BA.1, BA,2, BA.3, and a recently

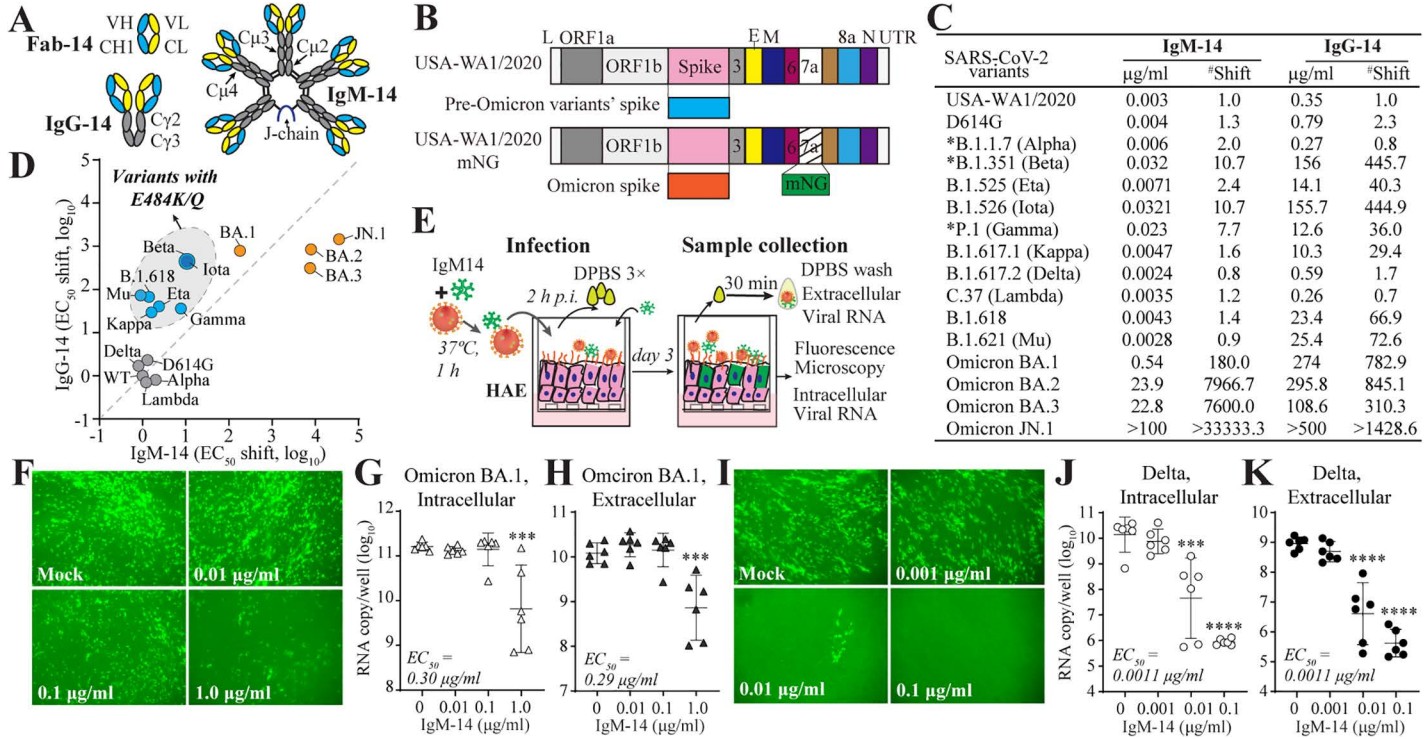

**Fig 1. Antibody neutralization activity and breadth against SARS-CoV-2. A**, Illustration of the architectures of Fab, IgG-14, and IgM-14. **B**, Construction of spike variants. L, 5' lead sequence; ORF, open reading frame; E, Envelope; M, Membrane; N, nucleocapsid; UTR: untranslated region; mNG, mNeonGreen. **C**, Summary of neutralization activities of IgM-14 and IgG-14 against naturally emerged variants. *, data from our previous study. #, Fold changes in $EC_{50}$ relative to the parental USA-WA1/2020 strain. **D**, Scatter plot comparing $EC_{50}$ fold changes of IgM-14 and IgG-14 across SARS-CoV-2 variants, normalized to the parental strain. Each point represents a variant. Blue, variants carrying the E484K/Q mutation; gray, variants without the E484K/Q mutation; orange, Omicron strains. **E**, Scheme of testing IgM-14's antiviral activity in human airway epithelial (HAE). **F-K**, antiviral efficacy of IgM-14 against SARS-CoV-2 in HAE cultures. Fluorescent microscopy of HAE infected with mNG BA.1-spike (**F**) or Delta-spike (**I**) variants after IgM-14 treatment. Intracellular viral RNA on day 3 after mNG BA.1-spike (**G**) or Delta-spike (**J**) infection. Extracellular viral RNA on day 3 after mNG BA.1-spike (**H**) or Delta-spike (**K**) infection. Geometric means with 95% confidence intervals (CI) are shown in panels **G-H** and **J-K**. One-way ANOVA with Dunn's multiple comparison corrections was used for statistical analysis. ***, $p < 0.001$; ****, $p < 0.0001$.

emerged JN.1. Pre-Omicron variants, which harbor single or a few spike mutations, were engineered into the ancestral USA-WA1/2020 SARS-CoV-2 strain (WT). Omicron variants, containing over 30 spike amino acid changes, were cloned into a live-attenuated USA-WA1/2020 strain expressing an mNG green fluorescence reporter [15] (Fig 1B and S1 Table). Neutralization activity was assessed using the plaque reduction neutralization test (PRNT) or PRNT-equivalent fluorescence focus reduction neutralization tests (FFRNT) [16].

IgM-14 potently neutralized all pre-Omicron variants, with half-maximal effective concentration ($EC_{50}$) values ranging from 0.003 to 0.0321 µg/ml, whereas IgG-14 showed significantly lower potency, with $EC_{50}$ in the range of 0.26-156 µg/ml (Figs 1C and S1). The E484K/Q mutation in the spike of pre-Omicron variants, a well-known antibody resistance marker [13], reduced IgG-14's neutralization potency by 29.4- to 445.7-fold (Fig 1C and 1D). In contrast, IgM-14 remained less affected by this mutation. It effectively neutralized Eta, Kappa, Mu, and B.1.618-variants carrying E484K/Q, with similar efficiency as against the ancestral strain and variants lacking this mutation, such as Delta, D614G, Alpha, and Lambda. The most significant reduction in IgM-14's neutralization (10.7-fold) was observed against Beta and Iota variants, which contained E484K and other RBD or NTD mutations (Fig 1C and 1D and S1 Table). Omicron variants become more resistant to both IgG-14 and IgM-14. IgG-14 was almost inactive against all tested Omicron sublineages ($EC_{50} > 100$ µg/ml).

IgM-14 showed modest neutralization against BA.1 ($EC_{50}$ of 0.54 µg/ml) and weak activity against BA.2 and BA.3 ($EC_{50}$s of 22.8-23.9 µg/ml) but was inactive against the Omicron sublineage JN.1 at the highest tested antibody concentration (Fig 1C and 1D).

Consistently, IgM-14 retained modest activity against BA.1 infection in physiologically relevant primary human airway epithelial (HAE) cell cultures. At a concentration of 1.0 µg/ml, IgM-14 markedly reduced the number of mNG-positive cells (Fig 1E and 1F). The estimated $EC_{50}$ for BA.1 inhibition in HAE ranged from 0.29 to 0.30 µg/ml, based on measurements of extracellular or intracellular viral RNA, respectively (Fig 1G and 1H). As a control, IgM-14 demonstrated substantially greater potency against the Delta variant, effectively reducing mNG-positive cells even at 0.01 µg/ml (Fig 1I). The $EC_{50}$ for Delta inhibition in HAE was 0.0011 µg/ml for both extracellular and intracellular viral RNA quantifications (Fig 1J and 1K). This represented a 263- to 273-fold decrease in neutralization efficacy against BA.1 compared to Delta.

Collectively, these findings further validate the broader neutralization spectrum of IgM-14 compared to IgG-14 against SARS-CoV-2 variants, and demonstrate the significant immune evasion of Omicron variants from IgM-14-mediated neutralization.

## IgM-14 resistance mutations map to the receptor-binding motif within the spike

To map residues critical for IgM-14 neutralization, we conducted serial passaging of live-attenuated mNG USA-WA1/2020 strain in the presence of increasing concentrations of IgM-14 (Fig 2A), following a protocol previously used to select IgG-14-resistant variants [12]. After six passages, three out of four independent selections (SL1–3) yielded variants strongly resistant to IgM-14, with $EC_{50}$ values exceeding 52 µg/ml, representing a > 17,000-fold increase in potency compared to the unpassaged parental strain.

Notably, the development of SARS-CoV-2 resistance to IgM-14 required a longer selection period (22 days versus 10 days for IgG-14) and more passages (6 versus 3 for IgG-14). Furthermore, one of the IgM-14 selections failed to yield resistant mutants, whereas all four IgG-14 selection experiments successfully recovered resistant variants (Fig 2A and our previous data [12]). These data support the notion that IgM-14 imposes a higher barrier to resistance compared to IgG-14.

Sequencing the spike gene from Passage 6 (P6) viruses revealed consensus amino acid changes, including G476D (in SL1 and SL3), F486S (in SL2), and two additional mutations near the furin cleavage site, S686G (in SL1), and Δ675–679 (deletion and the intact sequences co-existing in SL2 and SL3) (Figs 2B, S2A and S2B). G476D and F486S are located within RBM, a region frequently targeted by potent monoclonal antibodies [17,18]. The S686G and Δ675–679 were known Vero cell-adaptive changes that reduce S1 and S2 cleavage [19–21], and are unlikely to contribute to resistance. To validate the role of G476D and F486S in resistance to IgM-14, we introduced each mutation individually into the infectious clone of mNG SARS-CoV-2. Mutant viruses were recovered from VeroE6 cells following electroporation with *in vitro*-transcribed full-length RNA (Fig 2C). FFRNT analysis showed that IgM-14 failed to neutralize either G476D or F486S at concentrations up to 100 µg/ml (Fig 2D and 2E). Moreover, both mutant viruses exhibited complete resistance to IgG-14, confirming the critical roles of G476 and F486 in mediating neutralization by IgM-14 and IgG-14.

## Spike G476D and F486S mutations reduce SARS-CoV-2 fitness

Although both G476D and F486S mutant viruses replicated efficiently on VeroE6 cells, reaching titers exceeding $10^6$ PFU/ml, they formed smaller plaques compared to WT mNG-SARS-CoV-2 (Fig 2F), suggesting attenuated viral replication. To further assess the impact of both mutations on viral fitness, we compared their multicycle growth with WT mNG SARS-CoV-2 in immortalized and primary cell culture models. Both mutants exhibited varying degrees of replication attenuation across VeroE6, A549-hACE2, and HAE cultures (Fig 2G and 2H), demonstrating that these two mutations reduce viral fitness.

To understand how G476D and F486S mediate antibody resistance and impact viral fitness, we expressed, purified, and characterized the recombinant RBDs carrying each mutation. Interestingly, while F486S had only a marginal effect,

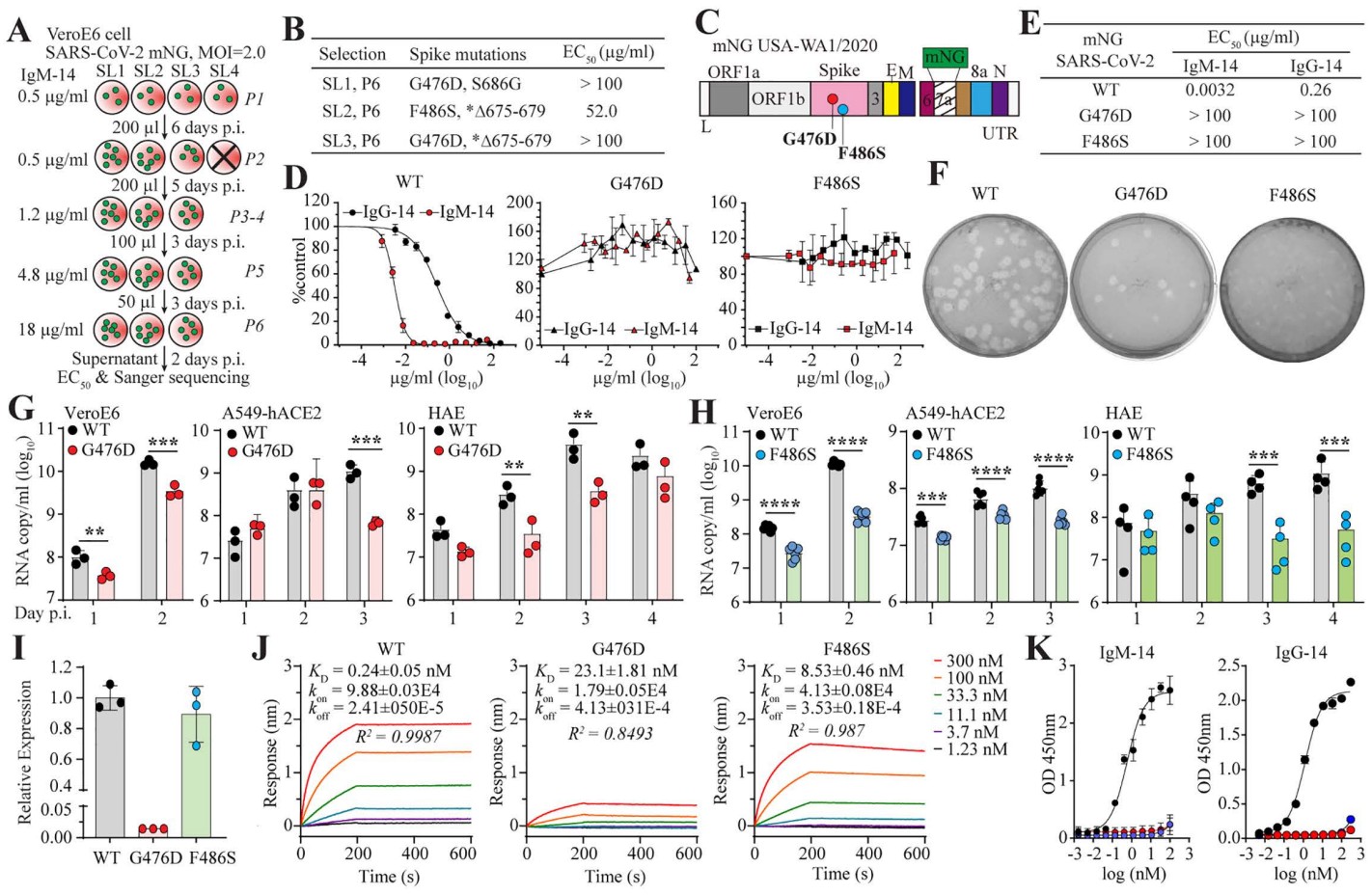

**Fig 2. Characterization of IgM-14-resistant variants. A**, Diagram of the process for selecting IgM-14-resistance in VeroE6 cells. Four independent selections (SL1-4) started at passage 1 (P1) and were passaged 6 times. SL4 was aborted at the second passage. **B**, Spike mutations occurred in P6. $EC_{50}$s of IgM-14 against P6 viruses (SL1-3) are shown. *, intact and deletion sequences observed in the selection. **C**, Construction of mNG SARS-CoV-2 with spike mutations. **D**, Neutralization curves of IgM-14 and IgG-14 against WT or mutant mNG SARS-CoV-2. **E**, Summary of the $EC_{50}$ values derived from panel **D. F**, Plaque morphologies of WT or mutant mNG SARS-CoV-2 on VeroE6 cells. Growth kinetics of G476D (**G**) or F486S (**H**) versus WT mNG SARS-CoV-2 in VeroE6, A549-hACE2, and human airway epithelial (HAE) cultures. Two-way ANOVA with Šidák's multiple comparison corrections was used for statistical analysis. **, $p < 0.01$; ***, $p < 0.001$; ****, $p < 0.0001$. **I**, Relative expression of RBD mutants to the WT RBD. Error bars indicate variation across three independent experiments. **J**, BLI analysis of the binding of WT and mutant RBDs to human ACE2. Association rate ($k_{on}$), dissociation rate ($k_{off}$), affinity constant ($K_D$), and $R^2$ values of curve fitting are indicated. **K**, ELISA analysis of IgM-14 and IgG-14 binding to WT (solid circles), G476D (red circles), and F486S (blue circles) RBDs. Error bars indicate variation across three independent experiments.

the G476D mutation drastically reduced RBD expression by approximately 71-fold (Fig 2I). This substantial reduction is possible due to the replacement of a neutral glycine with a negatively charged aspartic acid, which may disrupt RBD folding and compromise protein stability. The binding affinities of G476D and F486S RBDs to human ACE2, as determined by biolayer interferometry (BLI) assay, were 23.1 nM and 8.53 nM, respectively, which were 96- and 36-fold lower than that of the WT RBD (Fig 2J). In enzyme-linked immunosorbent assay (ELISA), neither mutant RBD bound to IgM-14 or IgG-14 (Fig 2K), confirming their role in antibody resistance. These findings demonstrate that IgM-14 targets functionally critical regions of the spike protein involved in receptor binding, and that resistance mutations in this region come at a significant cost to viral fitness.

## Structural heterogeneity of the spike/Fab-14 complex

We employed structural approaches to gain deeper insights into the molecular mechanism of IgM-14-mediated neutralization and resistance from naturally occurring and laboratory-acquired variants. Despite multiple attempts, no condition permitting cryo-EM imaging of isolated spike/IgM-14 complexes was found, likely due to the high flexibility and exceptionally strong avidity of IgM-14, which promoted extensive cross-linking and aggregation. Supporting this hypothesis, size-exclusion chromatography (SEC) of spike: IgM-14 (3:1) showed a prominent near-void peak (~8.3 ml; indicative of species exceeding the column's separation upper limit ~5,000,000 Da) and a broad 8.3-12.6 ml plateau, accompanied by a reduced IgM-14 peak (~12.6 ml), near disappearance of the spike peak (~13.8 ml), and minor late species >15 ml (S3A Fig). These features indicate polydisperse, very high-molecular-weight assemblies arising from multivalent cross-linking. As a control, spike: IgG-14 (2:3) resolved a single, discrete complex at ~12.5 ml and no near-void material (S3B Fig), consistent with a well-behaved, non-aggregated complex. In concert with the SEC, negative-stain electron microscopy revealed spike aggregation in the presence of IgM-14, distinct from the "starfish-like" morphology of IgM or the "mushroom-like" shape of the prefusion trimeric spike alone (S3C-S3E Fig). Notably, elongated, needle-like postfusion spike particles were also observed, plausibly arising from IgM-14-driven S1 dissociation. The appearance of later, smaller SEC species appeared compatible with such S1 release and subunit rearrangement. Together, these results indicated that IgM-14 promotes spike cross-linking/aggregation and conformational heterogeneity.

To overcome these technical challenges, we utilized Fab-14, which retains the key antigen-binding region of IgM-14 but lacks the Fc domain. This approach enabled us to successfully determine high-resolution cryo-EM structures of Fab-14 bound to the trimeric spike of the USA-WA1/2020 strain carrying the D614G mutation (hereafter referred to as the D614G spike).

We first established our cryo-EM imaging and reconstruction pipeline by determining the structure of the apo (Fab-free) D614G spike [22,23]. This initial dataset yielded relatively low-resolution cryo-EM maps that were nevertheless sufficient to reveal the overall spike architecture and confirm the correctness of our image reconstruction procedure (S4A-S4C Fig). The maps showed that RBDs undergo a hinge-like motion between up (receptor-accessible) and down (receptor-inaccessible) states, consistent with previous studies [24,25]. All well-defined spike trimers fell into two major conformational classes with roughly equal populations: one with a single RBD in the up state (referred to as "1-up-RBD"), and the other with all three RBDs in the down state (referred to as "3-down-RBD") (S4C Fig).

Next, we obtained high-resolution structures of Fab-14-bound D614G spike trimers after processing 17,415 micrograph movies followed by 2D classification, 3D reconstruction, and refinement (S5A-S5C Fig). The experiments were conducted at a Fab-14 to spike molar ratio of 5:1. Cryo-EM analysis showed that most spike trimers are bound by one or two Fab-14 molecules, hereafter referred to as "1-Fab" and "2-Fab" complexes. Only 7.2% of spike trimers remain unbound, all adopting the 3-down-RBD conformation (S5C Fig). According to the number of bound Fab-14s and conformational state of the spike, we identified five distinct modes of Fab-14/spike complexes (Figs 3A-3E and S5C).

Mode I (2-down-RBD/1-Fab) accounted for 21.4% of the total particles. In this mode, Fab-14 bridges two adjacent down RBDs in a bipartite manner with the third RBD being poorly resolved due to its local mobility (Figs 3A and S6A). Fab-14 binding appears to disrupt the threefold symmetry of three down RBDs by inducing a modest increase in the up angle of one of the two Fab-bound RBDs, and that of the Fab-free RBD. This reorganization is accompanied by positional shifts, with the centers of mass of the RBDs displaced by 1.2 to 3.0 Å (S6B-S6E Fig), relative to the published apo trimeric spike structures [26].

Mode II (1-up-RBD/1-Fab), which constitutes 26.3% of the total particles, features a single Fab-14 binding to an up RBD (Fig 3B). This mode is structurally heterogeneous and can be further divided into three subgroups with slightly different up angles (110°, 114°, and 117°) of the up RBD (S7A-S7B Fig).

Mode III (1-up-RBD/2-Fab) contains 11.4% of the total particles. In this mode, one Fab-14 molecule binds to an up RBD. At the same time, a second Fab-14 engages the other two down RBDs in a bipartite manner (Fig 3C). The

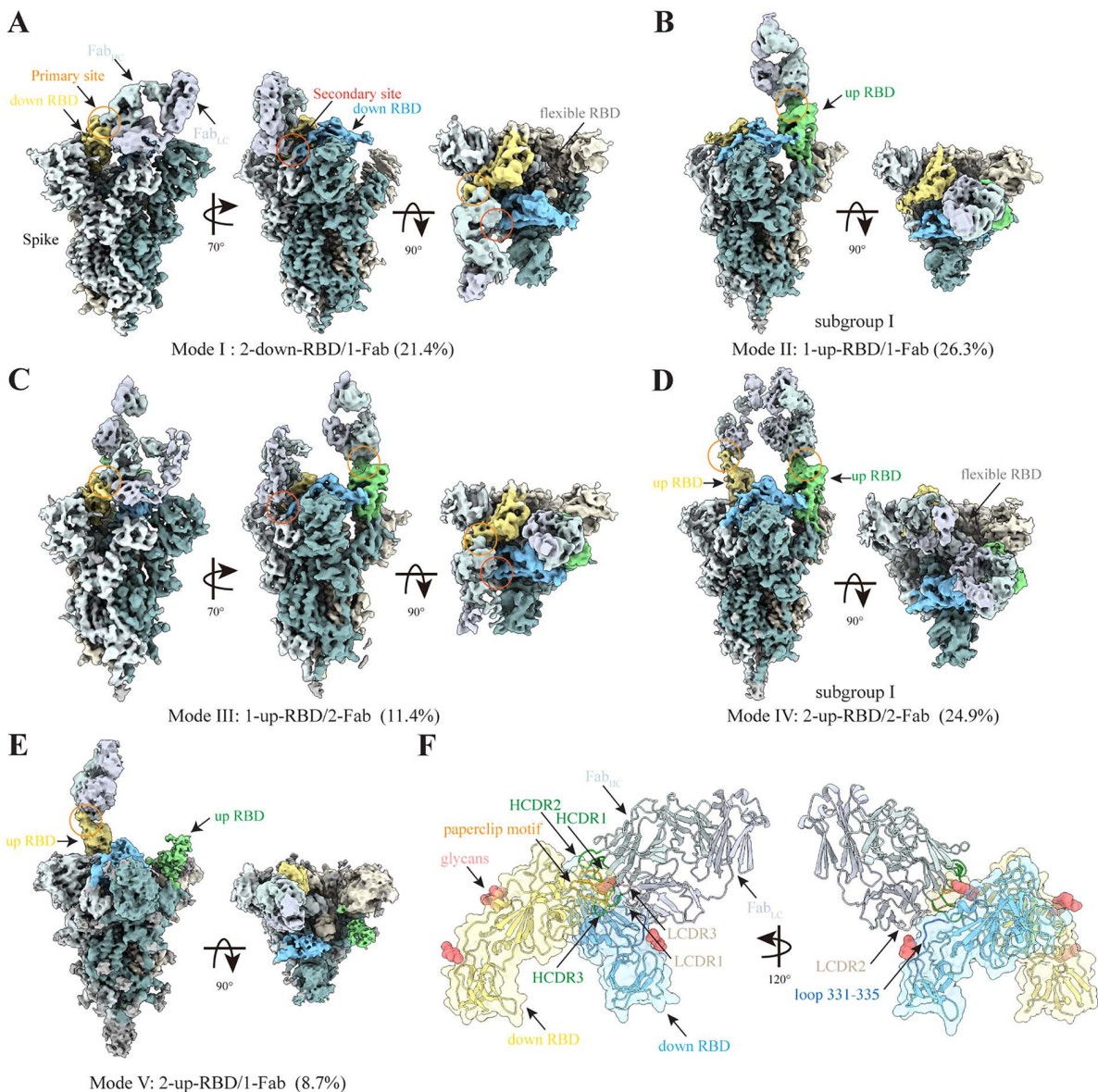

**Fig 3. Five modes of Fab-14 binding to D614G spike trimer. A,** Cryo-EM density of the Fab-14 and spike trimer complex in Mode **I.** The primary and secondary binding sites are indicated by orange and red circles, respectively. The down RBD involved in the primary binding site is shown in yellow, while the down RBD involved in the secondary binding site is shown in blue. The heavy and light chains of Fab-14 are colored pale cyan and light gray, respectively. **B,** Cryo-EM density of the Fab-14 and spike trimer complex in Mode **II.** The two down RBDs are shown in yellow and blue, respectively. The up RBD is shown in green. **C,** Cryo-EM density of the Fab-14 and spike trimer complex in Mode **III. D,** Cryo-EM density of the Fab-14 and spike trimer complex in Mode **IV.** The up RBDs are shown in yellow and green, respectively. The down RBD is shown in blue. **E,** Cryo-EM density of the Fab-14 and spike trimer complex in Mode **V. F,** local refined structures of primary and secondary binding sites in the bipartite binding mode. The paperclip motif in the primary binding site is shown in orange. The loop 331-335 in the secondary binding site is shown in dark blue.

orientation of the Fab-bound up RBD is similar to that found in Mode II, whereas the second Fab-14 binds the two down RBDs in a bipartite manner, similar to Mode I. Notably, to accommodate both Fab molecules, the up angles and position of the two down RBDs in this mode (S8A and S8B Fig) show slight differences compared to those in Mode I (S6B Fig).

Mode IV (2-up-RBD/2-Fab) was observed in 24.9% of the total particles. In this mode, two up RBDs are each bound by a Fab-14 (Fig 3D). Similar to Mode II, this group is heterogeneous. It can be divided into two subgroups (I and II), which differ by the orientation of their Fab-bound RBDs (S9A-S9D Fig) and by the amount of disorder in the down RBD (S5C Fig). In both subgroups, the two Fabs interact with each other via their light chains (S9C Fig).

Mode V (2-up-RBDs/1-Fab) was found in 8.7% of the total particles. In this mode, only one of the two up RBDs is bound by Fab-14 (Fig 3E). The Fab-bound up RBD density is significantly stronger than the free up RBD (S9D Fig). 3D flexible refinement confirmed conformational heterogeneity (S9E-S9F Fig), suggesting that Fab-14 binding may stabilize the RBD in the up conformation.

## Two distinct binding sites between Fab-14 and RBD

In all conformations of the spike/Fab-14 complex described above, the Fab-14/RBD unit displayed only two poses. In Modes II, III, IV, and V, Fab-14 binds to a single up RBD, while in Modes I and III, it bridges two adjacent down RBDs. To elucidate these interactions in atomic detail, we next performed local refinement of the Fab-14/RBD region using particle subsets that yielded the highest-quality EM density maps for Mode I and Mode IV subgroup I conformations.

The Mode I cryo-EM map refinement converged to a resolution of 3.4 Å (S5D Fig). In this conformation, Fab-14 engages two down RBDs via two distinct binding sites (Figs 3F, S5F and S5G). The primary site involves the paperclip motif (residues 471–491) within RBM, a region commonly targeted by Class 2 neutralizing antibodies [27,28]. Fab-14 interacts with this site via two heavy-chain complementarity-determining regions (HCDR-2 and HCDR-3) and two light-chain CDRs (LCDR-1 and LCDR-3). The secondary site comprises RBD residues 331–335, located near the hinge region and outside of the RBM, and is engaged with LCDR-2 of Fab-14.

Local refinement of the Mode IV subgroup I conformation revealed that two Fab-14 molecules engage two up RBDs through the paperclip motif as observed in Mode I (S5E, S5H and S10A Figs). Notably, the two Fabs also interact with each other via a light chain loop (residues 13–17) connecting strands A and B (S10B Fig). Structural alignment of the RBD/Fab-14 module at the primary binding site from Mode IV and Mode I showed a high similarity, with an RMSD of 0.21 Å across all Cα atoms (S10C Fig). These observations suggest that the structure of the RBD/Fab-14 module and at the primary binding interface are conserved across conformational modes. Consequently, since the secondary binding site involves a rigid-body rotation of the RBD/Fab-14 module (Fig 3F), it is also likely to be conserved.

## The primary site plays a major role in antibody neutralization

The Fab-14 heavy chain dominates the primary site interaction, with HCDR2 and HCDR3 residues forming a tight network of hydrophobic and polar contacts around the RBD paperclip motif (Fig 4A). Specifically, residues S57 and Y60 of the heavy chain coordinated E484. In contrast, Y54, L106, Y110, and Y112 of the heavy chain, along with Y93 of the light chain, form a hydrophobic clamp that captures F486 of the RBD (Fig 4B). F486 is sandwiched within this clamp by π-stacking interactions with R52 and Q104 of the heavy chain, flanking sides of its benzene ring. Notably, R52 is stabilized by ionic bridges with D62 and E102 of the heavy chain, while Q104 forms hydrogen bonds with R56 and Y110 of the heavy chain, as well as with RBD residue Y489. Y110 contributes to additional polar interactions with RBD N487. The light chain provides auxiliary stabilization through LCDR1 and LCDR3, with residues Y32, Y34, and Y93 engaging the loop spanning RBD residues 476–479 (Fig 4A and 4C). Collectively, Fab-14 buries a surface patch formed by RBD residues 456, 475–479, and 484–489, encompassing key determinants of ACE2 recognition. This primary binding site overlaps with the ACE2 interfaces (Fig 4D), suggesting that Fab-14 neutralizes the virus by directly blocking ACE2 engagement.

This structural model provides a framework for understanding the resistance mechanisms associated with several critical RBD mutations, G476D, E484K/A/S, and F486S/P, identified in natural variants. These mutations likely perturb this interaction network at the primary binding site, weakening the neutralization activity of IgM-14 and IgG-14. G476D can reduce backbone flexibility, introduce steric hindrance, and alter surface charge (Fig 4C). Substitution of E484 with alanine

 

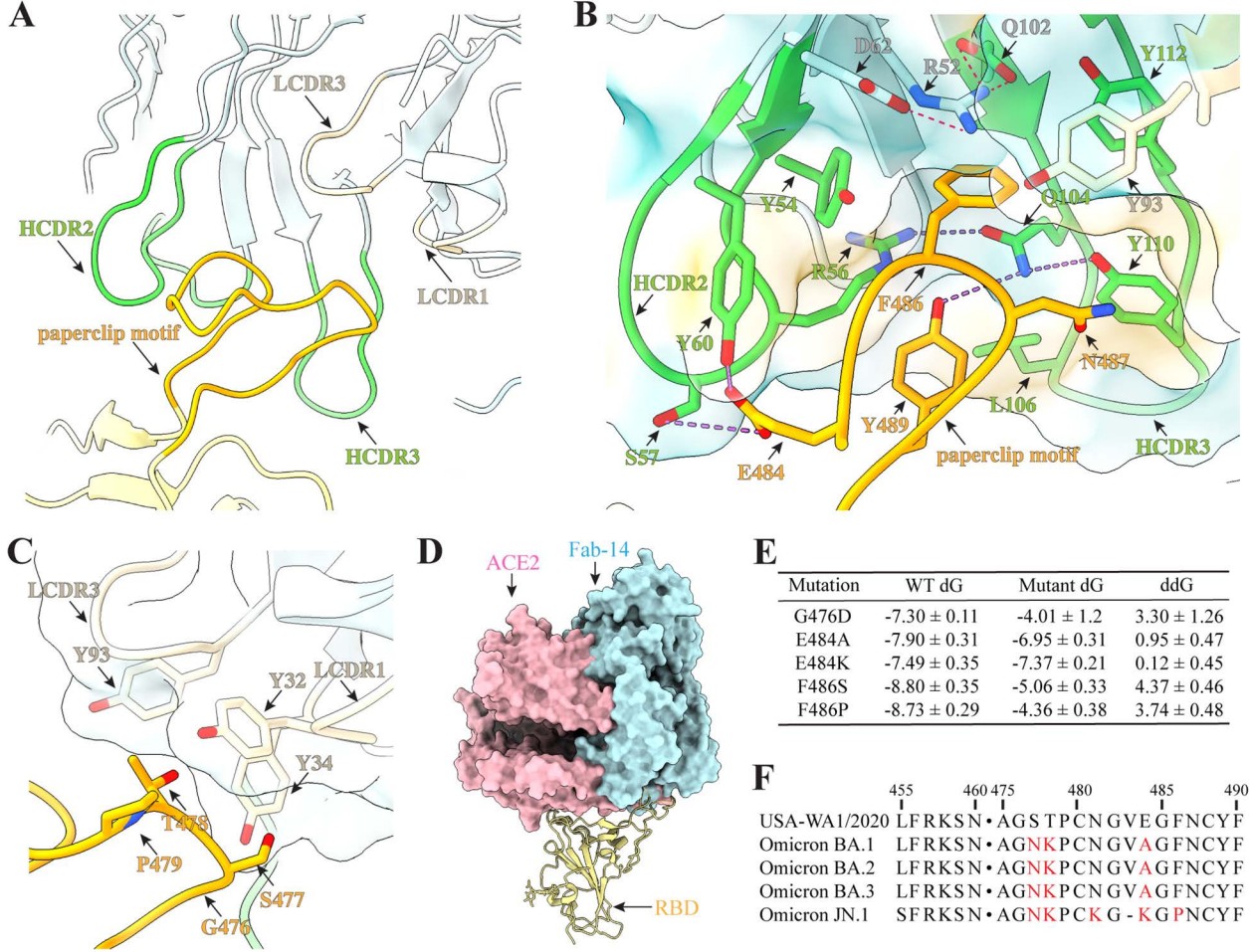

**Fig 4. Structural analysis of the primary binding site. A,** overview of the primary binding site. **B,** Zoomed-in view of the interaction between HCDR2 and the RBD paperclip motif. Pink dashed lines indicate salt bridges, purple dashed lines show hydrogen bonds. **C,** Zoomed-in view of the interaction between Fab-14 light chain and RBD residues G476-P479. **D,** Superposition of Fab-14 and ACE-2 on the same up RBD. The surface of ACE2 and Fab-14 are shown in pink and blue, respectively. **E,** MM/PBSA analysis of changes in RBD/Fab-14 complex binding free energy changes caused by individual mutations. Data shows the mean ± standard deviations. **F,** Sequence alignment of the primary binding site between the parental SARS-CoV-2 (USA-WA1/2020) and Omicron variants. The GISAIDs of BA.1, BA.2, BA.3, and JN.1 spikes were EPI_ISL_6640916, EPI_ISL_6795834.2, EPI_ISL_7605591, and EPI_ISL _18237538, respectively.

(A), serine (S), or lysine (K) can disrupt the glutamic acid-mediated polar interaction network, destabilizing HCDR2 binding (Fig 4B). F486S introduces a smaller, hydrophilic side chain, while F486P may alter the backbone conformation of the entire paperclip motif, both significantly impairing hydrophobic interactions and the "sandwich" stacking between Fab-14 and RBD (Fig 4B).

These structural suppositions were, in general, supported by Molecular Mechanics Poisson-Boltzmann Surface Area (MM/PBSA) calculations. Mutations involving interface-buried residues, F486S, F486P, and G476D, showed reduced binding free energy of RBD/Fab-14 complexes, whereas mutations at the peripheral, solvent-exposed E484 resulted in only modest changes (Fig 4E). Omicron variants carrying multiple mutations at the primary binding site, such as BA.1/BA.2/BA.3 (S477N, T478K, and E484A/K), and JN.1 (S477N, T478K, V483_del, E484K, and F486P) (Fig 4F), which likely further weaken Fab-14 binding.

## The secondary site contributes to IgM-14 neutralization

The local refinement structural analysis revealed that the second binding site exhibits fewer and weaker interactions, primarily concentrated in the 331–335 loop region of the RBD and Fab-14 LCDR2 (Fig 5A). Notably, the glycan attached to RBD residue N331 forms a stacking-like interaction with the side chain of LCDR2 residue Y62. Additionally, the backbones of RBD residues N331 and I332 interact with the side chain of Y62, while the Oγ atom of the T331 side chain forms hydrogen bonds with the backbone nitrogen atoms of LCDR2 residues G59 and I60. However, the relatively long distances (>3.9 Å) suggest these interactions are weak. Furthermore, the side chain of RBD residue L335 is 3.1 Å away from LCDR2 residue S58, indicating a van der Waals interaction. The glycan at RBD residue N343 interacts with the side chain

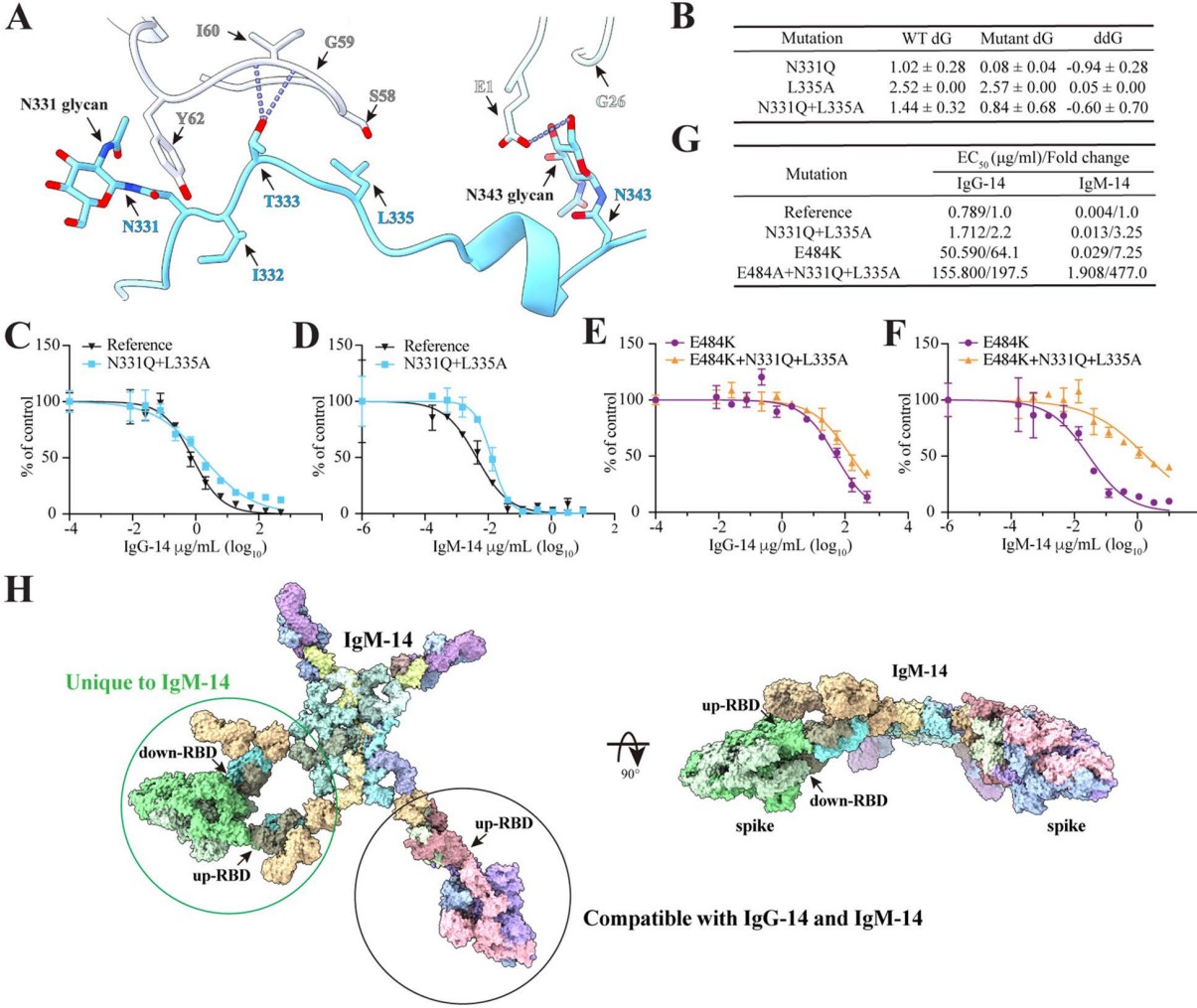

**Fig 5. Analysis of the secondary binding site. A**, Structural analysis of the secondary binding site. **B**, MM/PBSA analysis of changes in RBD/Fab-14 complex binding free energy changes caused by indicated mutations. **C-D**, Neutralization curves of IgG-14 (**C**) or IgM-14 (**D**) against mNG SARS-CoV-2 D614G (reference) and mNG SARS-CoV-2 D614G+N331Q+L335A (N331Q+L335A). **E-F**, Neutralization curves of IgG-14 (**E**) or IgM-14 (**F**) against mNG SARS-CoV-2 D614G+E484K (E484K) and mNG SARS-CoV-2 D614G+E484K+N331Q+L335A (E484K+N331Q+L335A). Means and standard deviation from three independent experiments are shown in panels **C-F**. **G**, Summary of EC$_{50}$ values of IgG-14 or IgM-14 against indicated SARS-CoV-2 mutants. **H**, A model of IgM-14 complexed with SARS-CoV-2 spike trimer shows different modes of spike-IgM-14 interactions. The coordinates of IgM were taken from a published report[41] by Chen, Q. et al. and adjusted by rigid body movement of the Fab arms while preserving the integrity of the Fab tethers.

of Fab-14 heavy chain residue E1 and the backbone of G26. However, the cryo-EM density in these regions indicated a high degree of conformational flexibility, suggesting that these interactions are not particularly stable.

To assess the biological significance of these secondary-site interactions, we designed three RBD mutants: N331Q, which removed the glycan-mediated interaction; L335A, which disrupted the hydrophobic contact; and the double mutant N331Q/L335A, which abolished both interactions. MM/PBSA analysis suggested that none of the mutations changed the binding free energy for the RBD/Fab-14 complexes (Fig 5B). To experimentally characterize the impact of the secondary site on antibody neutralization, we introduced the N331Q/L335A mutation into the infectious clone of a reference strain mNG SARS-CoV-2 D614G. FFRNT assays showed that the double mutant resulted in a modest 2.2- to 3.25-fold reduction in neutralization potency by IgM-14 and IgG-14 (Fig 5C, 5D and 5G), consistent with weak Fab-14 engagement at the secondary site. Consistent with these results, BLI analysis showed that Fab-14 binding to RBD fit a 1:1 interaction model across low, medium, and high RBD surface densities (S11 Fig). Increasing the RBD loading resulted in only minor changes in association rates and a slightly slower dissociation, with monophasic off-rates in all cases. These patterns showed that the contribution of the secondary contact to the overall energetics of Fab-14 binding is negligibly small compared to that of the primary binding site.

To further investigate the role of the secondary binding site, we introduced N331Q/L335A double mutation into the infectious clone of mNG SARS-CoV-2 D614G carrying the E484A mutation, which severely impaired IgG-14 binding at the primary site [12]. As expected, E484A reduced IgG-14's neutralization potency by 64.1-fold (EC$_{50}$ of 50.59 µg/ml for E484A versus 0.789 µg/ml for the reference strain), while IgM-14 was only modestly affected (7.25-fold increase in EC$_{50}$ values; 0.029 µg/ml versus 0.004 µg/ml). Notably, adding N331Q/L335A further reduced IgM-14 potency by 65.8-fold, compared to just a 3.08-fold reduction for IgG-14 (Fig 5E-5G). These data suggest that secondary-site interactions play a more critical role in IgM-14 neutralization than in IgG-14.

In addition to the structural analysis of Fab-14/D614G spike, we determined a cryo-EM structure of the BA.1 spike in the complex with Fab-14 (S12A-S12C Fig). Consistent with the modest neutralization potency of IgM-14, only a small fraction (11.1%) of BA.1 spike particles were bound by a single Fab-14 with all RBDs in the down conformation (S12C Fig). Although the map resolution was insufficient for atomic modeling, the overall architecture resembled Mode I of the Fab-14/G614 spike complex, where Fab-14 engages both primary and secondary binding sites (S13 Fig). This suggests that the conserved secondary binding site may partially compensate for the reduced affinity at the mutated primary site in BA.1. Consistently, these structural analyses highlight the critical role of the secondary site in IgM-14's neutralization breadth.

### Structural model of IgM-14 binding to SARS-CoV-2 spikes via both primary and secondary sites

To understand the structural basis for the role of the secondary binding site in IgM potency, we constructed multiple models of spike/IgM-14 interactions guided by the high-resolution structures of RBD/Fab-14 complexes. Because of its pentameric architecture and inherent structural flexibility, a single IgM-14 molecule can crosslink multiple spike trimers via diverse binding modes. As illustrated in Fig 5H, IgM-14 could engage a spike trimer via the binding Mode IV using a Fab pair from a single monomeric unit. More importantly, IgM-14 can bind a single spike via Modes I and II (one down-RBD and one up-RBD) using Fabs from adjacent monomeric units, with Mode I involving the secondary binding site. The remaining Fab arms remain available to bind additional spike trimers with all potential binding modes, enabling multivalent binding. In contrast, IgG-14, with its two Fab arms tethered to a single Fc hinge, is limited in angular flexibility and spatial reach, restricting it to binding one or two up RBDs or a single down RBD on a spike trimer. Overall, our structural model supports the enhanced avidity and broader spatial accessibility of IgM-14 compared to IgG-14, highlighting the functional importance of the secondary binding site in IgM-mediated neutralization.

### Discussion

In this study, we extend our previous findings [12,13] and demonstrate that pentameric engineered IgM-14 exhibits superior potency, a broader antiviral spectrum against SARS-CoV-2 variants, and a high barrier to resistance development

compared to its parental counterpart, IgG-14. Our structural analysis and molecular modeling provide insights into how IgM-14 engages with spike trimers through unique multi-mode mechanisms.

An antibody's epitope often determines its neutralization potency. Our Fab-14/spike trimer structure model shows that Fab-14 primarily targets a flap region within the RBM β-hairpin loop (residues A475 to Y489), consistent with prior epitope mapping [12]. This region overlaps with ACE2-contacts crucial for receptor engagement [29–32] and is commonly targeted by Class 2 antibodies, such as REGN10933 [18] and S2E12 [33,34] that potently neutralize pre-Omicron variants. Fab-14 engages this site through central hydrophobic interactions around RBD F486, along with peripheral contacts involving RBD residues E484 and Y489 on one side, and A475, G476, S477, and T478 on the other side (Fig 4). These interactions directly block ACE2 binding, which is the neutralization mechanism of IgG-14 and IgM-14. Mutations such as E484A, G476D, or F486S abolish IgG-14 or IgM-14 binding to purified RBDs (Fig 2I), validating the importance of both central and peripheral interactions in antibody binding affinity and their antiviral potency.

In addition to binding to the "up" RBD through the strong interaction at the primary site, Fab-14 bridges two neighboring "down" RBDs via bipartite interactions involving both primary and secondary sites (Fig 3). The secondary binding site, located outside of the RBM, forms fewer contacts with Fab-14 (Fig 5A). This bipartite binding mode appears to be unique to Fab-14; to our knowledge, this is the first report demonstrating the functional relevance of such a configuration in the IgM. However, no particles displaying only the secondary binding site were observed in the cryo-EM micrographs (S5C Fig). Mutations N331Q and L335A, which disrupt the secondary site, do not affect Fab-14/RBD binding energy and consistently have minimal impact on antibody neutralization (Fig 5B-5G). These findings showed that the contribution of the secondary contact to the overall energetics of Fab-14 binding is negligibly small compared to that of the primary binding site.

Spike conformational dynamics also significantly influence antibody neutralization [23,35]. The SARS-CoV-2 spike trimer adopts various conformations, with individual RBD swinging between "up" and "down" states [22,23,36]. While this structural heterogeneity is essential for ACE2 engagement, it also modulates epitope accessibility. Fab-14 targets a primary site accessible in both "up" and "down" RBD conformations. This distinguishes it from Class 1 antibodies, such as LY-CoV555 (bamlanivimab), which preferentially bind to "up" RBDs and are more vulnerable to escape when variants favor the "down" state [3537]. Cryo-EM analysis of Fab-14/spike trimer complexes revealed that Fab-14 binding induces distinct spike conformations, including "3-down", "1-up", and "2-up" (Fig 3A). Notably, the "2-up" spike trimer conformation (33.6% of particles) appears only upon Fab-14 binding, specifically in structural Modes IV and V (Figs 3A and S5D). However, such a conformation is absent in the apo spike structure (S4 Fig). Additionally, variations in RBD "up" angles Modes II and IV suggest Fab-14 stabilizes the RBD state. These findings indicate that Fab-14 not only binds across conformations but also reshapes the spike's conformational landscape.

Surprisingly, we observed substantial differences in IgG-14 neutralization escape among variants carrying similar primary site mutations (Fig 1C and S1 Table). For example, Iota, B.1.618, and Eta variants, all harboring the E484K mutation, exhibited markedly different $EC_{50}$ shifts: Iota> B.1.618 > Eta. Likewise, Beta, Mu, and Gamma variants, which share E484K and N501Y mutations along with additional RBD mutations (K417N/T or R346K), also showed distinct escape profiles: Beta > Mu > Gamma. These findings suggest that direct epitope alterations do not solely drive neutralization escape. Still, they may also involve mutations outside the binding site (such as NTD or S2 mutations) that influence spike conformation and/or epitope accessibility [38]. Further investigation is needed to elucidate the functional role of mutations outside the RBD in the context of specific variants.

Multivalency is a key factor contributing to IgM-14's superior neutralization potency and breadth compared to IgG-14. IgG-14, an IgG1 subtype, contains two Fab arms connected via a flexible hinge to a single Fc stem, allowing independent movement and binding to epitopes at varying orientations [39,40]. However, due to the limited length of Fab arms and steric constraints, IgG-14 typically binds to one or two RBDs per spike trimer (Fig 5H). In contrast, pentameric IgM carries 10 Fab arms attached to a rigid Fc core and the J-chain, which gives IgM a much larger spatial coverage. Cryo-EM studies

have shown that the hinge at the Cµ3/Cµ2 interface is flexible, allowing Fabs and Cµ2 to move and rotate [41], thus facilitating multivalent antigen binding.

IgM-14's multivalent architecture allows simultaneous engagement of spike trimers through diverse binding modes, regardless of the RBD's "up" or "down" conformation (Fig 5H). Beyond classical multivalent binding, IgM-14's unique ability to engage both primary and secondary sites introduces a non-canonical mechanism of avidity, contributing to its exceptional neutralization potency, breadth, and resistance tolerance. Even when primary-site affinity is compromised by mutations, such as E484K or E484A, IgM-14 retains substantial activity (Fig 1B). IgM-14, but not IgG-14, remains effective against the highly mutated BA.1 variant (containing S477N, T478K, and E484A at the primary binding site) in both VeroE6 and human primary cell cultures. Complete loss of neutralization is observed only when most critical primary-site interactions are abolished by mutations such as F486S, G476D (Fig 2), or E484K combined with F486P (Fig 1C and 1D), or when E484A is paired with secondary-site disruptions (N331Q and L335A) (Fig 5).

Although IgM-14's structural complexity presents challenges for high-resolution cryo-EM analysis, our alternative negative-stain EM analysis has revealed that IgM-14 may induce extensive spike aggregation and post-fusion-like particle formation (S3 Fig). This is likely driven by IgM-14's multivalency and high avidity. A similar phenomenon was observed with the AMETA nanobody platform, which used an IgM scaffold for multi-epitope targeting [42]. In addition to direct neutralization, IgM-14's agglutination capacity may further contribute to viral inactivation *in vivo* by immune cell-mediated clearance. Free IgM has been shown to adopt a planar conformation in which its ten Fab arms are evenly extended [43]. In this state, IgM engages its targets effectively, providing the structural rationale for the modeling shown in Fig 5H. Upon antigen binding, IgM transitions to a "staple" conformation that exposes its Fc domain more and promotes activation of the classical complement cascade [44].

In summary, our study elucidates the molecular basis of IgM-14's superior antiviral potency and breadth, highlighting the importance of multivalent binding and a non-canonical avidity mechanism through epitope-dependent engagement. This study provides valuable insights for developing ultra-potent, broad-spectrum anti-infective biologics, leveraging IgM scaffolds and other avidity-enhancing strategies such as IgA and bispecific antibodies [45].

## Materials and methods

### Cells and Virus

VeroE6 (ATCC CRL-1586) cells were obtained from the American Type Culture Collection (ATCC, Bethesda, MD). VeroE6 cells expressing TMPRSS2 (JCRB1819) were purchased from SEKISUI XenoTech, LLC. Both cells were cultured in Dulbecco's modified Eagle's medium (DMEM) supplemented with 10% fetal bovine serum (FBS; HyClone Laboratories, South Logan, UT) and 1% penicillin/streptomycin (P/S). All cultures were maintained at 37°C with 5% $CO_2$. A549-ACE2 cells (human alveolar epithelial cells expressing human Angiotensin-converting enzyme 2 [ACE2]) [46] were maintained in high-glucose DMEM supplemented with 10% FBS, 1% P/S, 1% 4-(2-hydroxyethyl)-1-piperazineethanesulfonic acid (HEPES), with 10 µg/ml blasticidin S. The 293F Freestyle cells (ThermoFisher Scientific) were maintained in FreeStyle 293 Expression Medium at 37°C with 8% $CO_2$. Human airway epithelia (HAE, EpiAirway, AIR-100) were obtained from MatTek Corporation and maintained according to the manufacturer's instructions. All antibiotics, medium, and supplements were purchased from ThermoFisher Scientific. All cells were tested Mycoplasma negative.

Infectious clone-derived SARS-CoV-2 and mNG SARS-CoV-2 were used in this study [47]. All SARS-CoV-2 handling was conducted at the Biosafety Level 3 facility with approval from the Institutional Biosafety Committee at the University of Texas Medical Branch.

### Selection of antibody-resistant viruses

VeroE6 cells were incubated with SARS-CoV-2 mNG (MOI 2) in the presence of IgM-14 at a starting concentration of 0.5 µg/ml. Supernatants were collected when more than 30% of cells showed CPE. 200 µl of supernatants were used for the

next infection on fresh VeroE6 cells with increased concentrations of IgM-14. The concentrations of IgM-14 in all 6 passages are the following: Passage 1–2 (P1-2): 0.5 μg/ml; P3-4: 1.2 μg/ml; P5: 4.8 μg/ml; P6: 18 μg/ml. The P6 viruses were used for testing their sensitivity to IgM-14 and for sequencing the spike region.

## PRNT

The neutralization of the SARS-CoV-2 US-WA1 strain and the recombinant SARS-CoV-2 variants were performed using the plaque reduction neutralization test (PRNT) as described previously [15]. In brief, antibodies were serially diluted in culture medium and incubated with 100 PFU of viruses at 37°C for 1 h, after which the antibody-virus mixtures were inoculated onto VeroE6 cell monolayers in six-well plates. After 1 h of infection at 37°C, 2 ml of 2% SeaPlaque agar (Lonza) in DMEM containing 2% FBS and 1% P/S was added to the cells. After 2 days of incubation, 2 ml of 2% SeaPlaque agar in DMEM containing 2% FBS, 1% P/S, and 0.01% neutral red (Sigma-Aldrich) was added on top of the first layer. After another 16 h of incubation at 37°C, plaque numbers were counted. The relative infection was obtained by the plaque counts from the antibody-treated groups compared to the untreated groups. The relative infection versus the concentration of the antibody (in $\log_{10}$ scale) was plotted. $PRNT_{50}$ titers were calculated using a nonlinear regression model in GraphPad software Prism 10.

## FFRNT

Fluorescent focus reduction neutralization test (FFRNT) was conducted using mNG SARS-CoV-2 by following a protocol described previously [16]. Briefly, VeroE6 cells were seeded in 96-well microplates (Greiner Bio-One). The cells were incubated overnight. The next day, 3-fold serially diluted IgM-14 and IgG-14 were mixed with each mNG SARS-CoV-2 at 37°C for 1 h. Afterward, the antibody-virus mixtures were loaded onto the pre-seeded VeroE6 cell monolayer. At 1 h post-infection, the inoculum was replaced with 100 μl of overlay medium (supplemented with 0.8% methylcellulose). The plates were incubated at 37°C for 16 h. Raw images of mNG foci were scanned and counted using Cytation 7 (BioTek) imager. The foci in each well were normalized to the non-antibody-treated controls to calculate the relative infectivities. The FFRNTs were performed in duplicate. A four-parameter nonlinear regression model in GraphPad Prism 10 determined the $EC_{50}$ values.

### Generation of SARS-CoV-2 mNG mutants

Recombinant mNG SARS-CoV-2 spike-variant was constructed by engineering the complete spike gene from the indicated variants into an infectious cDNA clone of mNG USA-WA1/2020 as reported previously [47]. Briefly, standard overlap PCRs using specific primers (listed in the S2 Table) were conducted to introduce the spike mutations into the infectious clone of the mNG USA-WA1/2020. The full-length infectious cDNA clones were assembled by *in vitro* ligation. Subsequently, the genome-length RNAs were synthesized by *in vitro* transcription. The full-length RNA and N gene RNA transcripts were electroporated into VeroE6-TMPRSS2 cells to rescue the recombinant viruses. After 48-72h of transfection, the supernatants (referred to as P0) were harvested. Then the P0 stocks were inoculated into freshly prepared VeroE6 cells for further amplification. At 48 h post-infection, supernatants (P1) were harvested and clarified with centrifugation at 1000 × g for 10 mins and stored at -80°C.

### RNA extraction, RT-PCR, and cDNA sequencing

Cell culture supernatants were mixed with a five-fold excess of TRIzol LS Reagent (ThermoFisher Scientific). Viral RNAs were extracted according to the manufacturer's instructions. The extracted RNAs were dissolved in 50 μl nuclease-free water. For sequence validation of mutant viruses, 2 μl of RNA samples were used for reverse transcription by using the SuperScript IV one-step RT-PCR kit (ThermoFisher Scientific) with Cov-21115V and Cov-25169R. The resulting DNAs

were purified by the QIAquick Gel Purification Kit and sequenced by Sanger sequencing at GENEWIZ (South Plainfield, NJ).

## Plaque assay

$8 \times 10^5$ VeroE6-TMPRSS2 cells per well were seeded into 6-well plates. The next day, 200 µl of 10-fold serially diluted virus was added to pre-seeded cells and incubated at 37°C for 1 h. After that, the inoculum was replaced with 2 ml of overlay medium containing DMEM with 2% FBS, 1% P/S, and 1% Seaplaque agarose (Lonza, Walkersville, MD, USA). After 2 days of incubation at 37°C, 2 ml of overlay medium supplemented with neutral red (Sigma-Aldrich) was added to stain the cells, and plaques were counted on the next day.

## Growth kinetics of recombinant SARS-CoV-2 in cell culture

For viral replication kinetics on VeroE6 and A549-hACE cells, $5 \times 10^5$ cells per well were seeded in 6-well plates. The next day, cells were infected with corresponding viruses (MOI 0.01 for VeroE6; MOI 0.1 for A549-hACE cells) at 37°C for 1 h. After infection, the inocula were removed, and the cells were washed with PBS three times. Cells were then incubated in DMEM supplemented with 2% FBS at 37°C, 5% $CO_2$. At each time point, 300 µl of supernatants were collected and 300 µl of fresh culture media were added back to the wells. All samples were stored at -80°C prior to use.

## Antiviral testing in HAE culture

For HAE (approximately $5 \times 10^5$ cells per well), an equal amount of SARS-CoV-2 mNG and variants (MOI 0.2) was added at the apical side and incubated at 37°C for 2 h. After infection, the inocula were removed, and the culture was washed three times with DPBS at the apical side and followed by incubation at 37°C, 5% $CO_2$. At each time point, 300 µl of DPBS was added onto the apical side of the airway culture and incubated at 37°C for 30 min to elute progeny viruses. All samples were collected from the apical side and stored at -80°C prior to use.
Antibodies at various concentrations were mixed with mNG SARS-CoV-2s and immediately used to infect HAE cultures (MOI 0.2). At day 3 post-infection, apical DPBS washes were collected as described above. Following DPBS wash collection, cells on the transwell inserts were lysed in 300 µl Trizol reagent (ThermoFisher Scientific). Viral RNAs in both cell lysates and DPBS washes were quantified by standard RT-qPCR. To determine the $EC_{50}$ values (the concentration of antibody required to inhibit 50% viral infection), relative RNA levels were calculated by normalizing viral RNA in each sample to those of no-antibody-treated control groups. A four-parameter nonlinear regression model in GraphPad Prism 10 was used to determine $EC_{50}$.

## RT-qPCR

To quantify viral yields, 1 volume of supernatant samples was mixed with 5 volumes TRIzol LS regent (ThermoFisher Scientific). SARS-CoV-2 RNA was extracted and eluted in 50 µl nuclease-free water according to the manufacturer's instructions. The RNA copies of each sample were determined by a standard curve-based quantitative RT-PCR (RT-qPCR) assay using iTaq SYBR Green One-Step Kit (BioRad) on the QuantStudio 7 Flex (ThermoFisher Scientific). An *in vitro* transcribed 3,839-bp RNA from the SARS-CoV-2 genome was used as a standard, as reported previously [48].

## Protein preparation

The mammalian expression plasmid pαH with cDNA encoding SARS-CoV-2 D614G spike HexaPro (S) was kindly provided by Dr. Jason S. McLellan [49]. A codon-optimized cDNA encoding the BA.1 spike sequence was synthesized by Genscript and cloned into the same pαH vector. Both constructs encode spike protein with HexaPro stabilizing mutations and a C-terminal $8 \times$ Histidine tag followed by $2 \times$ Strep-Tag II for purification purposes. D614G and BA.1 spike protein

were expressed and purified according by following a protocol reported previously [49]. In brief, FreeStyle 293-F cells were transfected with the plasmids encoding D614G spike or BA.1 spike using polyethyleneimine (Polysciences). After 3 h post-infection, 5 μM kifunensine (Sigma) was added to boost protein expression. On day 4 post-transfection, the supernatants were collected by centrifugation at 7000 × rpm at 4°C for 10 min, followed by passing through a 0.22 μm filter. The spike proteins were purified using Ni-NTA agarose columns (Cytiva) followed by size-exclusion chromatography using a Superose 6 10/300 column in an ÄKTA pure instrument (Cytiva). Spike proteins were finally eluted in a buffer composed of 2 mM Tris, pH 8.0, 200 mM NaCl.

The human ACE2 protein (10108-H08H) was purchased from Sino Biological. The Fc-tagged wild-type RBD protein and the RBD protein containing amino acid mutation F486S or G476D were prepared as described previously [45]. The IgG-14 was prepared as previously described [12]. Briefly, plasmids encoding the IgG-14 were transfected into Expi293F cells. 7 days post-transfection, the supernatants were collected and recombinant IgG-14 was purified using CaptivA HF Protein A Affinity Resin (REPLIGEN). Purified IgG-14 was then digested with papain to generate Fab-14 fragments. The IgM-14 was prepared as previously described [13]. Briefly, the $V_H$ and $V_L$ region of IgG-14 was incorporated into expression vectors encoding the IgM constant region and a human J-chain. IgM-14 was expressed in Expi293F cells and purified by mixed-mode chromatography and anion-exchange chromatography.

## ELISA

An Enzyme-Linked Immunosorbent Assay (ELISA) was performed to evaluate the antibody binding to the RBD proteins. The 96-well high-binding ELISA plates were coated with 100 μl per well of 0.2 μg/ml recombinant Fc-tagged RBD proteins overnight at 4°C. The plates were blocked with PBS supplemented with 5% skim milk. After blocking, 100 μl serial dilutions of monoclonal antibodies (diluted in 1% skim milk) were added to the wells and incubated at room temperature for 2 h. The plates were incubated with HRP-conjugated mouse anti-human lambda (Southern Biotech, 9280-05, 1:6,000 diluted in 1% skim milk) for 1 h. Between incubation steps, the plates were washed five times with PBST (0.05% Tween-20). After the final washes, TMB substrate was added at 100 μl per well for color development. The reaction was stopped by adding 50 μl per well 2M $H_2SO_4$. The $OD_{450}$ nm was read by a SpectraMax microplate reader and analyzed with GraphPad Prism 10.

## BLI

Bio-Layer Interferometry (BLI) was used to analyze RBD and ACE2 binding. The Fc-tagged RBD proteins were used for measuring affinity with human ACE2 on the Forte bio Octet RED 96 system (Sartorius, Goettingen, Germany). Briefly, RBD proteins (20 μg/ml) were captured onto protein A biosensors for 300 seconds. The loaded biosensors were then dipped into the kinetics buffer for 10 s for adjustment of baselines. Subsequently, the biosensors were dipped into serially diluted (0.37~300 nM) human ACE2 protein for 200 seconds to record association kinetics and then dipped into kinetics buffer for 400 s to record dissociation kinetics. A kinetic buffer without ACE2 was used to correct the background. The Octet Data Acquisition 9.0 software was used to collect affinity data. For the fitting of $K_D$ values, Octet Data Analysis software V11.1 was used to fit the curve by a 1:1 binding model and the use of the global fitting method. For RBD and Fab-14 binding, RBD was biotinylated with NHS-PEG4-Biotin (ThermoFisher) and captured on SA sensors (Sartorius) at 5, 20, or 100 μg/ml, titrated with Fab-14 (100-3.7 nM). Binding and dissociation were monitored in PBS, and data were globally fit using a 1:1 binding model with Octet Data Analysis software v13.

## Size-exclusion analysis

Purified D614G spike protein was incubated with IgM-14 or IgG-14 at a molar ratio of 3:1 and 2:3, respectively, at a final spike concentration of 0.2 mg/ml in a total volume of 200 μl. The mixture was incubated for 30 min at 4°C prior to analysis. Samples were then subjected to a Superose 6 Increase 10/300 column equilibrated in 2 mM Tris, pH 8.0, 200 mM NaCl.

## Cryo-EM sample preparation and imaging

For the D614G spike alone, 4 µL of proteins (0.3 mg/ml) were applied to the QUANTIFOIL R2/2 grid (Quantifoil) that had been plasma-cleaned by Solarus plasma cleaner (Gatan). The grid was plunge-frozen in liquid ethane using the EM GP2 Automatic plunge freezer (Leica) at 22°C and 95% humidity. For spike/Fab-14 complexes, Fab-14 was mixed with the D614G spike or BA.1 spike protein at a molecular ratio of 5:1 to a final concentration of 0.3 mg/ml spike and incubated at 4°C for 30 minutes. Then, 4 µL of samples were applied to the grid as described above.

Grids were loaded into a Titan Krios G3i microscope (ThermoFisher Scientific) equipped with a K3 or Falcon 4 direct electron detector with a GIF Quantum energy filter (20-eV energy slit) (Gatan) and operating at 300 keV. For the D614G spike alone, cryo-EM data were automatically acquired with a Falcon 4 camera using EPU software at a nominal magnification of 105,100 × (corresponding to 0.86 Å per pixel) with a nominal defocus range between -1.0 and -2.5 µm. Forty-frame movie stacks were collected over an exposure time of 1.0 s with a total dose of 40 e$^-$ Å$^{-1}$. For the D614G spike/Fab-14 complex, cryo-EM data were automatically acquired with a K3 camera using SerialEM at a nominal magnification of 105,100 × (corresponds to 0.84 Å per pixel) with a nominal defocus range between -1.0 and -2.5 µm. Forty-four-frame movie stacks were collected over an exposure time of 1.0 s with a total dose of 43.17 e$^-$ Å$^{-1}$. Similar movies were collected from frozen grids of the BA.1 spike/Fab-14 complex using the same microscope under the same conditions, except the stacks were collected over an exposure time of 1.0 s with a total dose of 43.27 e$^-$ Å$^{-1}$. The parameters of data collection were summarized in Supplementary S3-S6 Tables.

For the D614G spike alone, the collected movie fractions were imported into CryoSPARC for image processing [50]. Movie data were motion corrected using Patch Motion Correction. The contrast transfer function (CTF) was estimated using the Patch CTF Estimation method. Micrographs with CTF fit worse than 5 Å were excluded, leaving 10,220 micrographs. A total of 4,045,200 particles were selected using Blob Picker and extracted with 2 × binning (192 pixels). After four rounds of 2D classification, the Rebalance 2D classification was performed, and 317,242 particles of the total 2,095,808 particles were used to create five initial 3D volumes using Ab Initio with C3 symmetry. Three 3D volumes and all the remaining particles were used for five iterative rounds of heterogeneous refinement, resulting in two different conformations. Conformation 1 has one up-RBD with 419,193 particles. Conformation 2 has three down-RBD with 371,184 particles. The 419,193 particles in conformation 1 were re-extracted without binning (384 pixels) and further refined by homogeneous refinement, followed by global and local CTF refinement and a final round of non-uniform refinement with C1 symmetry, yielding a 3.3 Å reconstruction according to GSFSC at 0.143. The 371,184 particles in conformation 2 were re-extracted without binning (384 pixels) and further refined by homogeneous refinement with C3 symmetry, followed by global CTF refinement and a final round of non-uniform refinement with C3 symmetry, yielding a 3.3 Å reconstruction. The detailed information on data processing is shown in S4 Fig and S3 Table.

## Cryo-EM data processing

For the D614G spike/Fab-14 complex, collected movie fractions were imported into CryoSPARC and preprocessed similarly as described above. Micrographs with CTF fit worse than 4 Å were excluded, leaving 17,415 micrographs for further processing. A total of 7,261,652 particles were selected using Blob Picker and extracted with 2 × binning (220 pixels). After three rounds of two-dimensional (2D) classification, 695,411 remaining particles were used to create five initial three-dimensional (3D) volumes using Ab Initio. Three 3D volumes were used as input for three iterative rounds of heterogeneous refinement, resulting in various modes. Mode I contained 21.4% of the total particles. The 113,755 particles in Mode I were re-extracted without binning (440 pixels) and first refined using homogeneous refinement, followed by CryoSPARC's implementation of global CTF refinement and a final round of non-uniform refinement, yielding a 3.3 Å reconstruction. The particles were then refined with Flexible refinement. Local refinement was further performed in cryoSPARC to improve the electron density in the Fab-RBD interaction region. A mask that covers the Fab-14 and the two down RBD was created using UCSF Chimera. The mask was imported into cryoSPARC with dilation and soft padding.

Particles were re-centered to the center of the mask, and local refinement was performed, yielding a 3.4 Å reconstruction. Mode II contained 26.3% of the total particles. Heterogeneous refinement was conducted to further classify Mode II into three subgroups with different up angles. The particles were re-extracted and refined as described in Mode I, yielding 3.4 Å reconstruction for each subgroup. Mode III contained 11.4% of the total particles. The particles were re-extracted and refined as described in Mode I, yielding 3.3 Å resolution. Mode IV was observed in 24.9% of the total particles, which were divided into two subgroups using heterogeneous refinement. Local refinement was further performed in cryoSPARC to improve the electron density in the Fab-14/RBD contacts in Mode IV subgroup I. Two masks were created using UCSF Chimera: one covering the two 2-up-RBD and Fab-14, and one covering the remaining region. Masks were imported into cryoSPARC with dilation and soft padding. Particles were subtracted and re-centered, and local refinement was performed, yielding a 4.1 Å reconstruction. Mode V contained 8.7% of the total particles, which were re-extracted and refined as described in Mode I, yielding a 3.4 Å reconstruction. The remaining particles (7.2% of the total particles) are Fab-free, which were re-extracted and refined identically as Mode I, yielding a 3.4 Å reconstruction. Detailed information on data processing is shown in S5 Fig and S3 and S4 Tables.

For Omicron BA.1 spike/ Fab-14 complex, collected movie fractions were imported into CryoSPARC and preprocessed similarly to the D614G spike/Fab-14 complex. Micrographs with CTF fit worse than 4 Å were excluded, leaving 9,282 micrographs. A total of 1,123,585 particles were selected using Blob Picker and extracted with 2 × binning (192 pixels). After three rounds of 2D classification, 626,037 remaining particles were used to create five initial 3D volumes using Ab Initio. Three 3D volumes were used as input for three iterative rounds of heterogeneous refinement, resulting in three different conformations (1–3). The 336,659 particles in conformation 1 were re-extracted without binning (384 pixels) and further refined by homogeneous refinement, followed by CryoSPARC's implementation of global CTF refinement and a final round of non-uniform refinement, yielding a 3.1 Å reconstruction according to GSFSC at 0.143. The 105,694 particles in conformation 2 were re-extracted and refined as conformation 1, yielding a 3.2 Å reconstruction according to GSFSC at 0.143. The 55,801 particles in conformation 3 were re-extracted and refined identically as conformation 1, yielding a 3.6 Å reconstruction according to GSFSC at 0.143. The detailed information on data processing is shown in Supplementary S11 Fig and S6 Table. Local resolutions of final reconstructions were estimated by CryoSPARC's implementation of blocres and were colored by a resolution range from 3 Å to 7 Å in UCSF ChimeraX [50,51].

### Model building, refinement, and analysis

Cryo-EM structure of the D614G spike (PDB: 7BNM for 3-down-RBD; PDB: 7BNN for 1-up-RBD) and the atomic model of Fab-14 (generated by AlphaFold 3 [52]) were used to build an initial model by rigid body fitting in UCSF Chimera and manual adjustments in Coot. The model was refined iteratively by real-space refinement in Phenix and by manual adjustments and improvements in Coot. Protein-protein interaction was analyzed using UCSF ChimeraX and PyMol (Schrödinger). Figures were prepared using UCSF Chimera and UCSF ChimeraX. To calculate the angles of RBD-apo or RBD-Fab-14 complexes, the axes of RBDs were generated. Then, the angles between the horizontal plane and the axis were calculated with the UCSF Chimera X angle command [53].

### Molecular mechanics Poisson-Boltzmann Surface Area calculations

The MM/PBSA-derived binding free energies were calculated using FoldX package [54]. Firstly, the structure of a protein complex was refined by using the FoldX function "RepairPDB". This procedure "repairs" improbable dihedral angles and van der Waals clashes in a protein. The refined complex with respect to the FoldX energy function was then used as a starting point for a computational mutagenesis procedure with "BuildModel". A desired mutation was made to the repaired structure and then energy-minimized by sampling rotamers for the mutated residue and its neighboring amino acids. The wild-type protein was then energy-minimized with respect to rotamers of the same set of residues. The mutation-caused energy difference ($\Delta\Delta G$) between the mutant ($\Delta G_{MT}$) and wild type ($\Delta G_{WT}$) was calculated between the two chosen

protomers in the complex using the "AnalyseComplex" function. Apply the calculations for different pairs of protomers if the interaction involves more than two protomers. After repeating the calculations 10 times for each mutation, the means and standard deviations were calculated from structures with different local neighbor conformations.

## Negative-stain electron microscopy

For negative staining (NS) electron microscopy (EM), 4 µl of IgM-14 at 0.05 mg/ml was applied to a glow-discharged CF200-Cu carbon film grid (Electron Microscopy Sciences). The sample was left on the carbon film for 60 s, followed by negative staining with 2% uranyl formate for 60 s. Micrographs were recorded in a JEOL 2100FS microscope at 60,000 × magnification operated at 200 keV. The images were imported and processed in CryoSPARC. For the D614G spike alone, 4 µl of IgM-14 at 0.1 mg/ml was applied to the grid as described above. Micrographs were recorded and processed as described above, except images were recorded at 80,000 × magnification. For D614G spike and IgM-14 complex, D614G spike and IgM-14 were incubated on ice for 30 minutes at molar ratios of 6:1 or 3:1. The final concentration of D614S spike was kept at 0.1 mg/ml. 4 µl of complex was applied to the grid as described above, and micrographs were recorded and processed as the D614G spike alone.

## Statistics

Sample sizes were estimated based on similar research reported in the literature; no statistical method was used to predetermine sample size. No data was excluded from the analysis. All experiments were performed with at least two biological replicates to ensure reproducibility. Investigators were generally not blinded, as experimental conditions required knowledge of sample identities. Statistical tests can be found in the corresponding figure legends. Dots on the bar graphs represent the values from an individual replicate of the experiment. All data were analyzed using the software Prism 10 (GraphPad). Unless otherwise noted, numerical results from repeated independent experiments are presented as mean ± standard deviation (SD). For viral growth kinetics, data were $log_{10}$-transformed to approximate a normal distribution before statistical analysis.

## Supporting information

**S1 Fig. Neutralization curve of IgM-14 and IgG-14 against SARS-CoV-2 variants.** Red circles and curves represent response to IgM-14 treatment; open black circles with solid dark curves indicate response to IgG-14 treatment. USA-WA1/2020 has been tested for four different batches, each with three replicates. Means from three independent experiments are shown. Error bars indicate standard deviations.
(DOCX)

**S2 Fig. Representative Sanger sequencing chromatography of spike gene.** A, chromatography of sequences encoding spike amino acids 475–494. B, chromatography of sequences encoding spike amino acids 671–691. Mutations are highlighted.
(DOCX)

**S3 Fig. Negative-stain EM and size exclusion column analysis of D614G spike in complex with IgM-14.** A, SEC analysis of D614G spike mixed with IgM-14. The elution volume for each major peak is indicated. B, SEC analysis of D614G spike mixed with IgG-14. C, Representative negative-stain EM micrographs and 2D class averages of IgM-14 alone. Black circles indicate "starfish-like" IgM-14 particles. D, Representative micrographs and 2D classes of D614G spike alone. Blue circles indicate intact prefusion trimeric spike particles. E, Representative micrographs and 2D classes of D614G spike in the presence of IgM-14 at the indicated molar ratios. Red circles indicate postfusion-like spike particles. Scale bar, 100 nm.
(DOCX)

**S4 Fig. Cryo-EM data processing and reconstruction of D614G spike.** A, Representative cryo-EM micrograph of D614G spike alone. Scale bars, 100 nm. B, Representative 2D classes of the entire dataset. C, Diagram of cryo-EM data process. Two distinct conformations (1-up-RBD and 3-down-RBD) with the corresponding Gold Standard Fourier Shell Correlation (GSFSC) and local resolution estimation are shown.
(DOCX)

**S5 Fig. Cryo-EM data processing and reconstruction of D614G spike in complex with Fab-14.** A, Representative micrograph. Scale bars, 100 nm. B, Representative 2D class averages. C, Diagram of cryo-EM data process. Five modes (I-V) were shown. Three subgroups with slightly different up angles in Mode II and two subgroups with slightly different orientations of the Fab-bound RBDs in Mode IV are shown. D, Local refinement of complex Mode I. E, Local refinement of complex Mode IV subgroup I. GSFSC and local resolution estimation for each map are shown. F, Primary binding site from local refinement of Mode I, showing the fitted atomic model within the cryo-EM density. G, Secondary binding site from local refinement of Mode I, showing the fitted atomic model within the cryo-EM density. H, Primary binding site from local refinement of Mode IV, subgroup I, showing the fitted atomic model within the cryo-EM density.
(DOCX)

**S6 Fig. Protomer of D614 spike-Fab-14 in Mode I.** A, Superimposition of cryo-EM maps for Mode I and its 5 Å low-pass filtered maps. B, Angles between the axis of each RBD in Mode I and the horizontal plane are shown to the right of each chain. C, Distance between each down RBD in Mode I. D, Angles between the axes of RBDs in the D614 spike and the horizontal plane are shown to the right of each chain. E, Distance between the center of mass of each down RBD in D614G spike. D-E, A three-fold symmetric spike structure (PDB ID: 7bnm) was used for comparison.
(DOCX)

**S7 Fig. Analysis of up RBD in three subgroups of Mode II.** A, Alignment of cryo-EM maps of subgroups I-III. B, Angles between the axes of the up RBD (fitted into the cryo-EM maps in rigid bodies) in the three subgroups.
(DOCX)

**S8 Fig. Analysis of Mode III.** A, Angles between the axes of each RBD (fitted into the cryo-EM maps in rigid bodies) in three subgroups. B, Top view of Mode III. Distance between the centers of mass of two down RBDs.
(DOCX)

**S9 Fig. Analysis of Mode IV and Mode V.** A, Angles between the axes of each up RBD in Mode IV subgroup I. B, Angles between the axes of each up RBD in Mode IV subgroup II. C, Side view of the two up RBDs in Mode IV subgroup I and Mode IV subgroup II. The dashed line indicates the center distance between RBDs. D, Superimposition of cryo-EM maps for Mode V and its 10 Å low-pass filtered maps. E, Scatter plots illustrating the dispersion of particle latent coordinates throughout the Mode V dataset. F, A representative cryo-EM map of Mode V after 3D flexible refinement training.
(DOCX)

**S10 Fig. Structural analysis of the primary binding site.** A, Structural comparison of two up RBD-Fab-14 after local refinement. B, Interactions between two Fab-14s. C, Structural comparison of the primary binding site when Fab-14 is bound to a down-RBD and an up-RBD. The Fab-14 complexed with down-RBD is shown in gray, while Fab-14 complexed with up-RBD is shown in salmon.
(DOCX)

**S11 Fig. BLI analysis of Fab-14 binding to immobilized RBD at different surface densities.** RBD was immobilized on the sensor at 5 (top), 20 (middle), or 100 (bottom) µg/ml and titrated with Fab-14 at the indicated concentrations. Apparent kinetic parameters ($K_D$, $k_{on}$, and $k_{off}$) and fitting $R^2$ (1:1 binding) values derived from global fitting are indicated.
(DOCX)

**S12 Fig. Cryo-EM processing and validation of Omicron BA.1 Spike and Fab-14 complex.** A, Representative micrograph. Scale bars, 100 nm. B, Representative 2D classes. C, Diagram of cryo-EM data process. Three distinct conformations were resolved. Confirmation 1–2 represents Fab-free BA.1 spike protein, where conformation 1 contains one up-RBD and conformation 2 has three down-RBD. Confirmation 3 has one Fab-14 bind to two down-RBD. The GSFSC and local resolution estimation for each map were shown.
(DOCX)

**S13 Fig. Cryo-EM density map for the Fab-14/BA.1 spike complex.** The down RBD at the primary site is shown in yellow. The down RBD at the secondary site is shown in dark blue.
(DOCX)

**S1 Table. Spike mutations in SARS-CoV-2 variants.**
(DOCX)

**S2 Table. List of primers used in this study.**
(DOCX)

**S3 Table. Statistics for 3D reconstruction for D614G spike alone.**
(DOCX)

**S4 Table. Statistics for 3D reconstruction and model refinement of Fab-14/D614G spike complex.**
(DOCX)

**S5 Table. Statistics for 3D reconstruction and local refinement of Fab-14/D614G RBD structure.**
(DOCX)

**S6 Table. Statistics for 3D reconstruction of Fab-14/BA.1 spike complex.**
(DOCX)

**S1 Data. Raw data.**
(XLSX)

## Acknowledgments

We thank Jason S. McLellan and his team for providing the initial plasmid for SARS-CoV-2 spike expression and for their technical guidance during this project. We would like to acknowledge the Sealy Center for Structural Biology and Molecular Biophysics at the University of Texas Medical Branch (UTMB) for providing technical support and access to the cryo-EM facility.

## Author contributions

**Conceptualization:** Petr G. Leiman, Xuping Xie.

**Data curation:** Yan Wang, Yanping Hu, Zhiqiang Ku, Jason Yeung, Jing Zou, Michael Woodson, Nikolai S. Prokhorov, Ekaterina S. Knyazhanskaya, Haiqing Zhao, Michael B. Sherman, Petr G. Leiman, Xuping Xie.

**Funding acquisition:** Pei-Yong Shi, Petr G. Leiman, Xuping Xie.

**Investigation:** Yan Wang, Yanping Hu, Zhiqiang Ku, Jason Yeung, Jing Zou, Michael Woodson, Nikolai S. Prokhorov, Ekaterina S. Knyazhanskaya, Haiqing Zhao, Michael B. Sherman, Petr G. Leiman, Xuping Xie.

**Methodology:** Yan Wang, Yanping Hu, Zhiqiang Ku, Jason Yeung, Jing Zou, Michael Woodson, Nikolai S. Prokhorov, Ekaterina S. Knyazhanskaya, Haiqing Zhao, Michael B. Sherman, Petr G. Leiman, Xuping Xie.

**Resources:** Zhiqiang An, Stephen F. Carroll, Petr G. Leiman, Xuping Xie.

**Supervision:** Petr G. Leiman, Xuping Xie.

**Writing – original draft:** Yan Wang, Yanping Hu, Petr G. Leiman, Xuping Xie.

**Writing – review & editing:** Yan Wang, Yanping Hu, Zhiqiang Ku, Jason Yeung, Jing Zou, Michael Woodson, Nikolai S. Prokhorov, Ekaterina S. Knyazhanskaya, Haiqing Zhao, Michael B. Sherman, Zhiqiang An, Stephen F. Carroll, Pei-Yong Shi, Petr G. Leiman, Xuping Xie.

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
