## [Decision Letter · Decision Letter 0]

15 Jan 2026

PPATHOGENS-D-25-03146

Neutralization of SARS-CoV-2 by IgM-14 via Engagement of Two Distinct Spike Epitopes

PLOS Pathogens

Dear Dr. Xie,

Thank you for submitting your manuscript to PLOS Pathogens. After careful consideration, we feel that it has merit but does not fully meet PLOS Pathogens's publication criteria as it currently stands. Therefore, we invite you to submit a revised version of the manuscript that addresses the points raised during the review process.

We look forward to receiving your revised manuscript.

Kind regards,

Chee Wah Tan

Academic Editor

PLOS Pathogens

Ashley St. John

Section Editor

PLOS Pathogens

Sumita Bhaduri-McIntosh

Editor-in-Chief

PLOS Pathogens

orcid.org/0000-0003-2946-9497

Michael Malim

Editor-in-Chief

PLOS Pathogens

orcid.org/0000-0002-7699-2064

**Journal Requirements:**

At this stage, the following Authors/Authors require contributions: Yan Wang, Yanping Hu, Zhiqiang Ku, Jason Yeung, Jing Zou, Michael Woodson, Nikolai S. Prokhorov, Ekaterina S. Knyazhanskaya, Haiqing Zhao, Michael B. Sherman, Zhiqiang An, Stephen F. Carroll, Pei-Yong Shi, Xuping Xie, and Petr G. Leiman. Please ensure that the full contributions of each author are acknowledged in the "Add/Edit/Remove Authors" section of our submission form.

https://journals.plos.org/plospathogens/s/submission-guidelines#loc-parts-of-a-submission

4) We do not publish any copyright or trademark symbols that usually accompany proprietary names, eg ©,  ®, or TM  (e.g. next to drug or reagent names). Therefore please remove all instances of trademark/copyright symbols throughout the text, including:

- ® on page: 18 and 22.

5) Please upload all main figures as separate Figure files in .tif or .eps format. For more information about how to convert and format your figure files please see our guidelines:

6) We have noticed that you have uploaded Supporting Information files, but you have not included a list of legends. Please add a full list of legends for your Supporting Information files after the references list.

7) Please amend your detailed Financial Disclosure statement. This is published with the article. It must therefore be completed in full sentences and contain the exact wording you wish to be published.

State what role the funders took in the study. If the funders had no role in your study, please state: "The funders had no role in study design, data collection and analysis, decision to publish, or preparation of the manuscript.".

**Reviewers' Comments:**

Reviewer's Responses to Questions

**Part I - Summary**

Reviewer #1: In this manuscript, Wang et al. investigates the neutralization activity of a previously characterized SARS-CoV-2 IgM antibody, IgM-14, against newly emerged variants and presents cryo-EM structures of Fab-Spike complexes. By extending earlier functional observations to additional variants and providing structural information, the manuscript offers useful insights into the neutralizing mechanism of the IgM-14. The study is technically sound, the data are of good quality, and the overall scope and depth of the work are well aligned with the standards of the journal. Providing more solid evidence and explanations in the following aspects would further strengthen the manuscript.

Reviewer #2: This study examines IgM-14, an engineered IgM antibody against SARS CoV 2, and explains why it is so potent and how viral variants escape it. They found IgM-14 is far more powerful than its IgG version against all pre-Omicron SARS‑CoV‑2 variants, but its activity drops sharply against Omicron BA.1 and is completely lost against the later JN.1 variant. Two mutations on Spike RBD, G476D and F486P, were found with resistance to the IgM-14. Further cryo-EM structural study revealed two binding interfaces on RBD, which explains why the IgM-14 showed enhanced potency than IgG-14. Overall the study is important and interesting, providing a new dual-engage model of antibody neutralization.

Reviewer #3: In the manuscript tilted “Neutralization of SARS-CoV-2 by IgM-14 via Engagement of Two Distinct Spike Epitopes” Wang, Hu et al., study the mechanism of SARS-CoV-2 neutralization by IgM-14, a therapeutic antibody that was administered intranasally. The main finding of the paper is that the IgM version of the antibody leverages the secondary contact that the Fab makes with the adjacent RBD. By determining cryo-EM structures of Fab-14 with spike, the authors have revealed high resolution details of the interaction. Overall, the paper reports important data on a clinically relevant antibody and provide a model for multivalent association with IgM.

The authors should address the following:

1. The NSEM data are not sufficient to draw the conclusion that IgM-14 promoted aggregation. The spike sample itself shows some clustering and the IgM+spike sample looks messier but still to make the conclusion about IgM-mediated aggregation, the authors should use a complementary measure such as light scattering or size exclusion chromatography.

2. The use of the MM/PBSA calculations seems underdeveloped and are not adding more insights beyond what can be observed directly with the structures. These calculations do not consider the impact of the mutations on the conformational landscape of the spike and are an oversimplification, therefore, likely to be misleading. I recommend removing this and focusing solely on the experimental structural and functional analysis.

3. In the supplement, authors should show overlays of map and model with the map contoured appropriately to demonstrate resolution. This should be done for different regions of the structures, especially the Fab/spike interactive regions.

4. In the section covered in lines 279-287, the authors refer to distances and hydrogen bonds and main chain interactions, but none of these are shown in the figure (5A). Authors should revise the figure.

5. Line 347-348 – Other antibodies are known that make peripheral contact with the adjacent RBD or the adjacent NTD. What this paper shows that these secondary contacts can be functionally relevant. Authors should consider revising/modulating this statement.

6. For the BLI data, authors should show the fit to the curves overlayed on the sensorgrams. For some of the interactions (for example, 2J WT), there is almost nothing falling off the sensor tip in the off phase, therefore, how accurate/robust are the affinity and kinetics values provided?

Minor:

Line 187: Typo “(Fi S4A-C)”

Figure 1A, legend. I think the authors mean “Illustration” and not “Illusion”

**Part II – Major Issues: Key Experiments Required for Acceptance**

Please use this section to detail the key new experiments or modifications of existing experiments that should be absolutely required to validate study conclusions.required to validate study conclusions.required to validate study conclusions.required to validate study conclusions.

Reviewer #1: 1. The interaction between Fab-14 and two down RBDs is described as involving two interfaces: a primary interaction with residues 471-491 within the RBM and a secondary interaction with residues 331-335 of the neighboring down RBD. While the primary RBM interaction is well supported, the evidence for the secondary interface is comparatively weak. Specifically, the manuscript does not provide sufficiently resolved cryo-EM density at this interface to clearly support side-chain interactions or glycan involvement, nor are complementary binding assays (such as BLI assays using mutant RBD proteins) presented to directly validate this interaction. Although a slight reduction in neutralization activity is observed for the mutants incorporated with E484K, this alone does not convincingly establish a critical functional role for the proposed secondary site in the S-Fab-14 interaction. Moreover, the formation of such a secondary contact is, to some extent, understandable given the intrinsic conformational flexibility of RBDs (e.g., up and down configurations) which underlying the apparent structural tolerance among the three RBDs within the trimer. In this context, statements emphasizing a critical role of the secondary site, for example, “Site-directed mutagenesis and structural modeling validated the critical role of this secondary site in IgM-14-mediated neutralization”, appear to be overstated and should be reconsidered or more carefully qualified.

2. The conclusion that IgM-14 can simultaneously engage Spike through multiple interfaces in diverse modes, thereby revealing a noncanonical avidity mechanism distinct from IgG-14, is an interesting hypothesis. However, this interaction model would benefit from more direct structural evidence at the IgM-Spike level. Cryo-EM or negative-stain EM data visualizing the intact IgM-14 in complex with Spike would substantially strengthen the claim of simultaneous engagement and better support the statements in the Abstract, such as “Unlike IgG-14, IgM-14 can simultaneously engage both interfaces in diverse modes” and “highlight the structural and functional uniqueness of IgM-14.”

Reviewer #2: 1. Since there are two distinct binding sites on RBD, did the author observe the secondary binding event in BLI assay when the Fab associated to the immobilized RBD? Usually the sensorgrams will display a heterogeneous binding mode for such a dual-binding event, which can be better fit with a bivalence model.

2. If the secondary binding event can be observed in BLI, what is the kon/koff and KD?

3. Is there any synergism between the two binding site?

4. Will the secondary binding site exclusively available in down conformation? Will there be clash or still compatible for the Fab binding to the site when RBD is up?

Reviewer #3: (No Response)

**Part III – Minor Issues: Editorial and Data Presentation Modifications**

Reviewer #1: Figure 1C: There appears to be no corresponding symbol “#” in the table, although it is mentioned in the legend. In addition, “Lota” should be corrected to “Iota.”

Line 199: The current naming of Mode V could be reconsidered. Among the five conformations described, the Spike structures can be broadly grouped into two categories: 1-up-RBD and 2-up-RBD states. Modes I-III all correspond to the 1-up-RBD conformation. Renaming Mode I as “1-up-RBD/1-Fab-1” and Mode II as “1-up-RBD/1-Fab-2,” for example, may help clarify the distinctions among these modes.

Line 247: Cryo-EM density maps showing side-chain features of the interface residues from both Fab-14 and RBD should be provided to support the detailed description of the interface and proposed interaction forces.

Line 280: Similarly, cryo-EM density maps of side chains and glycans at the interface should be shown to better support the detailed structural interpretation of the interactions.

Reviewer #2: 5. Minor point: label in Figure 3A seems wrong. The yellow ‘down RBD’ should be ‘up RBD’ instead.

Reviewer #3: (No Response)

PLOS authors have the option to publish the peer review history of their article (If published, this will include your full peer review and any attached files.). If published, this will include your full peer review and any attached files.

...

Reviewer #1: No

Reviewer #2: No

Reviewer #3: No

**Figure resubmission:**
---

## [Decision Letter · Decision Letter 1]

9 Mar 2026

Dear Dr. Xie,

We are pleased to inform you that your manuscript 'Neutralization of SARS-CoV-2 by IgM-14 via Engagement of Two Distinct Spike Epitopes' has been provisionally accepted for publication in PLOS Pathogens.

Best regards,

Chee Wah Tan

Academic Editor

PLOS Pathogens

Ashley St. John

Section Editor

PLOS Pathogens

Sumita Bhaduri-McIntosh

Editor-in-Chief

PLOS Pathogens

orcid.org/0000-0003-2946-9497

Michael Malim

Editor-in-Chief

PLOS Pathogens

orcid.org/0000-0002-7699-2064

Reviewer Comments (if any, and for reference):

Reviewer's Responses to Questions

**Part I - Summary**

Reviewer #1: (No Response)

Reviewer #2: The author has addressed my questions adequately. No further question from my side.

Reviewer #3: The authors have adequately addressed all my critiques.

**Part II – Major Issues: Key Experiments Required for Acceptance**

Please use this section to detail the key new experiments or modifications of existing experiments that should be absolutely required to validate study conclusions.required to validate study conclusions.required to validate study conclusions.required to validate study conclusions.

Reviewer #1: (No Response)

Reviewer #2: (No Response)

Reviewer #3: (No Response)

**Part III – Minor Issues: Editorial and Data Presentation Modifications**

Reviewer #1: (No Response)

Reviewer #2: (No Response)

Reviewer #3: (No Response)

...

Reviewer #1: No

Reviewer #2: No

Reviewer #3: No

---

## [Editor Report · Acceptance letter]

Dear Dr. Xie,

We are delighted to inform you that your manuscript, "Neutralization of SARS-CoV-2 by IgM-14 via Engagement of Two Distinct Spike Epitopes," has been formally accepted for publication in PLOS Pathogens.

Best regards,

Sumita Bhaduri-McIntosh

Editor-in-Chief

PLOS Pathogens

orcid.org/0000-0003-2946-9497

Michael Malim

Editor-in-Chief

PLOS Pathogens

orcid.org/0000-0002-7699-2064